# Social stratification without genetic differentiation at the site of Kulubnarti in Christian Period Nubia

Kendra A. Sirak [1,2,3,4✉], Daniel M. Fernandes [4,5,6], Mark Lipson[1,2], Swapan Mallick[1,7,8], Matthew Mah [1,7,8], Iñigo Olalde [1,9], Harald Ringbauer [1,2], Nadin Rohland[1,7], Carla S. Hadden[10], Éadaoin Harney[1,2,11], Nicole Adamski[1,8], Rebecca Bernardos [1], Nasreen Broomandkhoshbacht [1,8,16], Kimberly Callan [1,8], Matthew Ferry[1,8], Ann Marie Lawson [1,8,17], Megan Michel[1,8,18], Jonas Oppenheimer[1,8,19], Kristin Stewardson[1,8], Fatma Zalzala[1,8], Nick Patterson[7,20], Ron Pinhasi [4,5,20], Jessica C. Thompson [3,12,14,15,20], Dennis Van Gerven[13,20] & David Reich [1,2,7,8,20]

Relatively little is known about Nubia's genetic landscape prior to the influence of the Islamic migrations that began in the late 1st millennium CE. Here, we increase the number of ancient individuals with genome-level data from the Nile Valley from three to 69, reporting data for 66 individuals from two cemeteries at the Christian Period (~650–1000 CE) site of Kulubnarti, where multiple lines of evidence suggest social stratification. The Kulubnarti Nubians had ~43% Nilotic-related ancestry (individual variation between ~36–54%) with the remaining ancestry consistent with being introduced through Egypt and ultimately deriving from an ancestry pool like that found in the Bronze and Iron Age Levant. The Kulubnarti gene pool – shaped over a millennium – harbors disproportionately female-associated West Eurasian-related ancestry. Genetic similarity among individuals from the two cemeteries supports a hypothesis of social division without genetic distinction. Seven pairs of inter-cemetery relatives suggest fluidity between cemetery groups. Present-day Nubians are not directly descended from the Kulubnarti Nubians, attesting to additional genetic input since the Christian Period.

[1] Department of Genetics, Harvard Medical School, Boston, MA 02115, USA. [2] Department of Human Evolutionary Biology, Harvard University, Cambridge, MA 02138, USA. [3] Department of Anthropology, Emory University, Atlanta, GA 30322, USA. [4] Earth Institute and School of Archaeology, University College Dublin, Dublin 4, Ireland. [5] Department of Evolutionary Anthropology, University of Vienna, Vienna 1090, Austria. [6] CIAS, Department of Life Sciences, University of Coimbra, 3000-456 Coimbra, Portugal. [7] Broad Institute of Harvard and MIT, Cambridge, MA 02142, USA. [8] Howard Hughes Medical Institute, Harvard Medical School, Boston, MA 02115, USA. [9] Institute of Evolutionary Biology, CSIC-Universitat Pompeu Fabra, Barcelona, Spain. [10] Center for Applied Isotope Studies, University of Georgia, Athens, GA 30602, USA. [11] Department of Organismic and Evolutionary Biology, Harvard University, Cambridge, MA 02138, USA. [12] Department of Anthropology, Yale University, New Haven, CT 06511, USA. [13] Department of Anthropology, University of Colorado at Boulder, Boulder, CO 80309, USA. [14] Yale Peabody Museum of Natural History, New Haven, CT 06511, USA. [15] Institute of Human Origins, Arizona State University, Tempe, AZ 85287, USA. [16]Present address: Department of Anthropology, University of California, Santa Cruz, CA 95064, USA. [17]Present address: Department of Human Genetics, University of Michigan Medical School, Ann Arbor, MI 48109, USA. [18]Present address: Department of Human Evolutionary Biology, Harvard University, Cambridge, MA 02138, USA. [19]Present address: Department of Biomolecular Engineering, University of California, Santa Cruz, CA 95064, USA. [20]These authors jointly supervised this work: Nick Patterson, Ron Pinhasi, Jessica C. Thompson, Dennis Van Gerven, David Reich. ✉email: kendra_sirak@hms.harvard.edu

Situated along the Nile River between the First Cataract at Aswan in present-day Egypt and the confluence of the Blue and White Nile Rivers near the present-day Sudanese capital of Khartoum (Fig. 1a), Nubia has a long and dynamic history of continual human occupation and is a place where people from multiple parts of Africa and West Eurasia interacted[1–5]. Throughout the 20th century, archeological expeditions studied the relationships between Nubian groups and people from both north of the Sahara Desert and sub-Saharan Africa, informing debates about the history of this region[6–14]. Archeological and historical evidence attest to a particularly dynamic relationship between Nubia and Egypt established more than 6000 years ago that increased in intensity over time. The introduction of Christianity beginning in 542 CE[15–18] and the Arab conquests that began in Egypt in the 7th century CE and expanded southward along the Nile over the next 700 years[1,19,20] reflect West Eurasian influence in Nubia, with Egypt often serving as intermediary.

Especially when integrated with archeology, ancient DNA can provide insight into the processes that shaped the genomes of ancient populations. Here we present genome-wide data from 66 individuals who lived at Kulubnarti, located between the Second and Third Cataracts of the Nile approximately 120 km south of the Sudanese city of Wadi Halfa, during the earlier part of the Christian Period (~650–1000 CE). Kulubnarti represents an ideal context in which to investigate the genetic ancestry of Nubians in

the mid- to late-1st millennium CE and the study of its Christian Period inhabitants provides a unique opportunity to shed light on fine-scale questions raised by archeological and bioarcheological research.

Kulubnarti is located in the desolate *Batn el Hajar* ("belly of rock") region separating Lower Nubia (the northern part of Nubia between the First and Second Cataracts) and Upper Nubia (the southern part of Nubia). Studies of cranial and dental traits suggest that the Kulubnarti Nubians were similar to ancient people from Wadi Halfa, located to the north near the Nile's Second Cataract[8,9]; however, morphological data have limited resolution for determining biological relationships relative to genome-wide data. Genetic studies of present-day Nubians reveal a mix of sub-Saharan African- and West Eurasian-related ancestry, but the mixture is largely a result of the Arab conquest of the late-1st and early-2nd millennia CE[20], a time during which people with West Eurasian-related ancestry spread southward along the Nile through Egypt and into Nubia[1,4]. Because more recent admixture events obscure our understanding of the ancestry of people who predate these events, the analysis of paleogenomic data from Kulubnarti offers an opportunity to directly investigate the ancestry and biological relationships of a Nubian group that lived in the region before the introduction of Islam.

Ancient DNA analysis of the Kulubnarti Nubians also provides an opportunity to resolve fine-scale questions about the

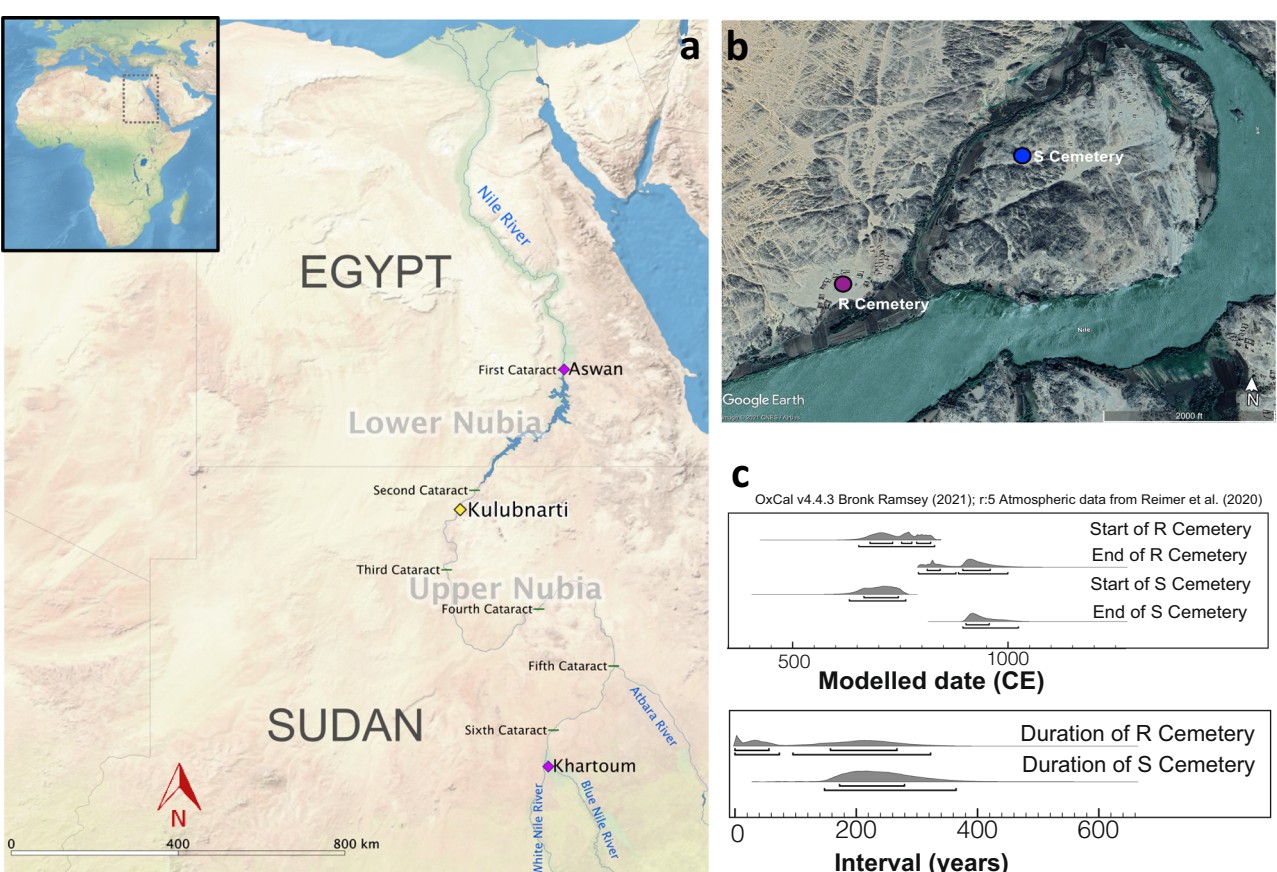

**Fig. 1 Geographic and temporal context of Kulubnarti. a** Map of Nubia, including location of Kulubnarti (yellow diamond), present-day cities of Aswan and Khartoum (magenta diamonds), and the six numbered cataracts of the Nile River (green lines). Inset map shows the location of focus within Africa. **b** Zoomed view of Kulubnarti showing the location of sites 21-S-46 ('S cemetery') and 21-R-2 ('R cemetery') marked with yellow diamonds. Maps made with QGIS Geographic Information System v.3.6.0 and Google Earth Pro; basemaps from Natural Earth (naturalearthdata.com) (**a**) and Google Earth (**b**). **c** Modeled start and end dates (top) and duration of use (bottom) of R and S cemeteries based on 29 newly generated radiocarbon (14C) dates shows their contemporaneity. Brackets indicate the 68.3% highest posterior density (hpd) and 95.4% hpd ranges, the former of which are referred to in the text (Supplementary Note 2; individual 14C data provided in Supplementary Table 1 and modeled in Supplementary Fig. 1).

relationships among the individuals who lived there. Archeologists excavated two cemeteries at Kulubnarti ~1 km apart, both with Christian-style burials[21]. Site 21-S-46 (the 'S cemetery') was situated near the west side of Kulubnarti Island (a true island only at the peak of the Nile flood), and site 21-R-2 (the 'R cemetery') was located on the mainland opposite the southern end of the island (Fig. 1b; Supplementary Note 1; Supplementary Fig. 1)[21]. Grave types and grave goods were indistinguishable between the cemeteries, but osteological analyses identified significant differences in morbidity and mortality using markers of generalized stress (e.g., cribra orbitalia)[22,23], patterns of growth and development[24,25], and average life expectancy[22]. On average, people buried in the S cemetery experienced more stress and disease and died younger than those buried in the R cemetery (Supplementary Note 1).

Similarities in grave styles, considered alongside differences in morbidity and mortality and archeological evidence suggesting that individuals from the R cemetery were of higher economic status, inspired the hypothesis that Kulubnarti was home to a culturally homogenous population divided into two socially-stratified groups that lived separately and utilized separate burial grounds[21-23,26]. This scenario draws not just on anthropological and archeological evidence from Kulubnarti, but is also inspired by ethnographic studies of recent Nubians, where sometimes semi-itinerant, landless, ethnically Nubian people live apart from, and provide occasional labor for, landowning Nubians. Under the proposed hypothesis, a similar socioeconomic structure may have existed during the Christian Period, whereby those individuals buried in the S cemetery provided seasonal labor for the landowning individuals who were buried in the R cemetery[26]. Before this study, the question of whether these cemeteries were the burial grounds of genetically distinct groups was unanswered. Paleogenomic data can elucidate whether systematic genetic differences accompanied the economic and social differences between the people buried in the two cemeteries.

Here we show that the Christian Period Kulubnarti Nubians were admixed with Nilotic-related and West Eurasian-related ancestry, the latter likely introduced into Nubia through Egypt but ultimately most like that found in the Levant during the Bronze and Iron Ages. The Kulubnarti gene pool was formed over the course of at least a millennium and shows evidence of limited population-level relatedness that implies connectivity with a broader population. These connections may have been female-mediated, as suggested by a finding of disproportionately female-associated West Eurasian-related ancestry; this provides a new line of evidence that Kulubnarti may have been a patrilocal society. The identification of inter-cemetery relatives is consistent with a scenario of fluidity between groups. Ancient DNA provides a new line of evidence supporting the hypothesis that the burial of people in two cemeteries at Kulubnarti was not strongly rooted in genetic differences.

## Results

**Ethics statement.** We acknowledge the ancient individuals whose remains we analyzed and who must be treated with respect. Excavation of human remains from the two Kulubnarti cemeteries occurred in 1979 under a license granted by the Sudan Antiquities Service (now the National Corporation for Antiquities and Museums) to Dr. William Y. Adams; the excavation of the cemeteries was funded by the National Science Foundation (Grant No. 77-270210-535), and led by Dr. Dennis Van Gerven, co-senior author on this work; it was undertaken as a part of the UNESCO International Campaign to Save the Monuments of Nubia. Prior to the excavation of the Kulubnarti cemeteries, the head of the Sudanese Antiquities Service approved the research plan, including the invasive investigations (such as biochemical

analyses) anticipated at the time. All graves to be excavated were marked by the archeological team and their excavation was approved by the Nubian *reiss* (foreman). The excavation was inspected by a representative of the Sudan Antiquities Service monthly, and all excavated remains were reinspected prior to their export, which occurred in accordance with the regulations at that time. At the time of excavation, although the people living around Kulubnarti did not identify remains in Christian-style graves as their ancestors and communicated this to Dr. Van Gerven and his colleagues, they lived in close geographic proximity to the site and continued to use the R cemetery as a burial ground for their deceased. Based on this connection, they provided community perspectives as the custodians of this cemetery and gave their consent for scientific work. Following the wishes of the local community, Dr. Van Gerven's team avoided any interference with Muslim graves in the Kulubnarti cemeteries, which were distinguishable by their north–south orientation.

**Data overview.** We screened 111 individuals from Kulubnarti for authentic ancient DNA (see the "Methods" section; Supplementary Data 1) and enriched promising libraries for sequences overlapping ~1.24 million genome-wide SNPs[27-30]. We obtained genome-wide data for 66 individuals (27 from the R cemetery and 39 from the S cemetery) with coverage averaging 0.28× at targeted positions (Supplementary Data 2). We analyzed these data jointly with sequences from published ancient African[31] and West Eurasian individuals[29,30,32-42] (Supplementary Data 3), as well as from present-day people living in Africa and West Eurasia[20,43-49].

We generated direct radiocarbon dates for 29 individuals (Supplementary Table 1; Supplementary Fig. 1) and constructed a Bayesian chronological model to estimate the start and end dates and duration of Christian-style burials in each cemetery (Supplementary Note 2). Archeological evidence from both cemeteries, including Christian-style burials identified by grave orientation, body positioning, and lack of associated grave goods, suggests contemporaneous use[21]. Our direct dates support contemporaneity (Fig. 1c). Christian-style burials in the R cemetery began 680–830 calibrated years CE (calCE) and continued for up to 270 years until 810–960 calCE (all modeled dates represent 68.3% highest posterior density [hpd]). Similarly, Christian-style burials in the S cemetery began 660–750 calCE and continued for up to 280 years until 900–960 calCE. The burials analyzed here therefore span the so-called Early Christian Period (550–800 CE) and the earlier part of the Classic Christian Period (850–1100 CE).

Thirty-three individuals from Kulubnarti (half of those studied here) had at least one and up to five genetic relatives in our dataset, sharing 28 pairwise genetic relationships that formed eight extended families (Fig. 2c; Supplementary Note 3; Supplementary Table 2). Four pairs were first-degree relatives (three from the S cemetery and one from the R cemetery), and we excluded the lower-coverage individual from each pair in group-level analyses (Supplementary Data 1). We document seven relative pairs (closest being second-degree relatives) where one individual was buried in the R cemetery and the other in the S cemetery; there is no age or sex pattern associated with the inter-cemetery relative burials. This observation provides a second line of evidence that the cemeteries were contemporaneous and reveals that close relatives were not always buried in the same place. Comparing whether 32 individuals with relatives of a known-degree in our dataset were part of within- or across-cemetery relative pairs, we find a significant reduction in the expected rate of cross-cemetery relatives if cemetery of burial exhibited no correlation to family structure when we pool

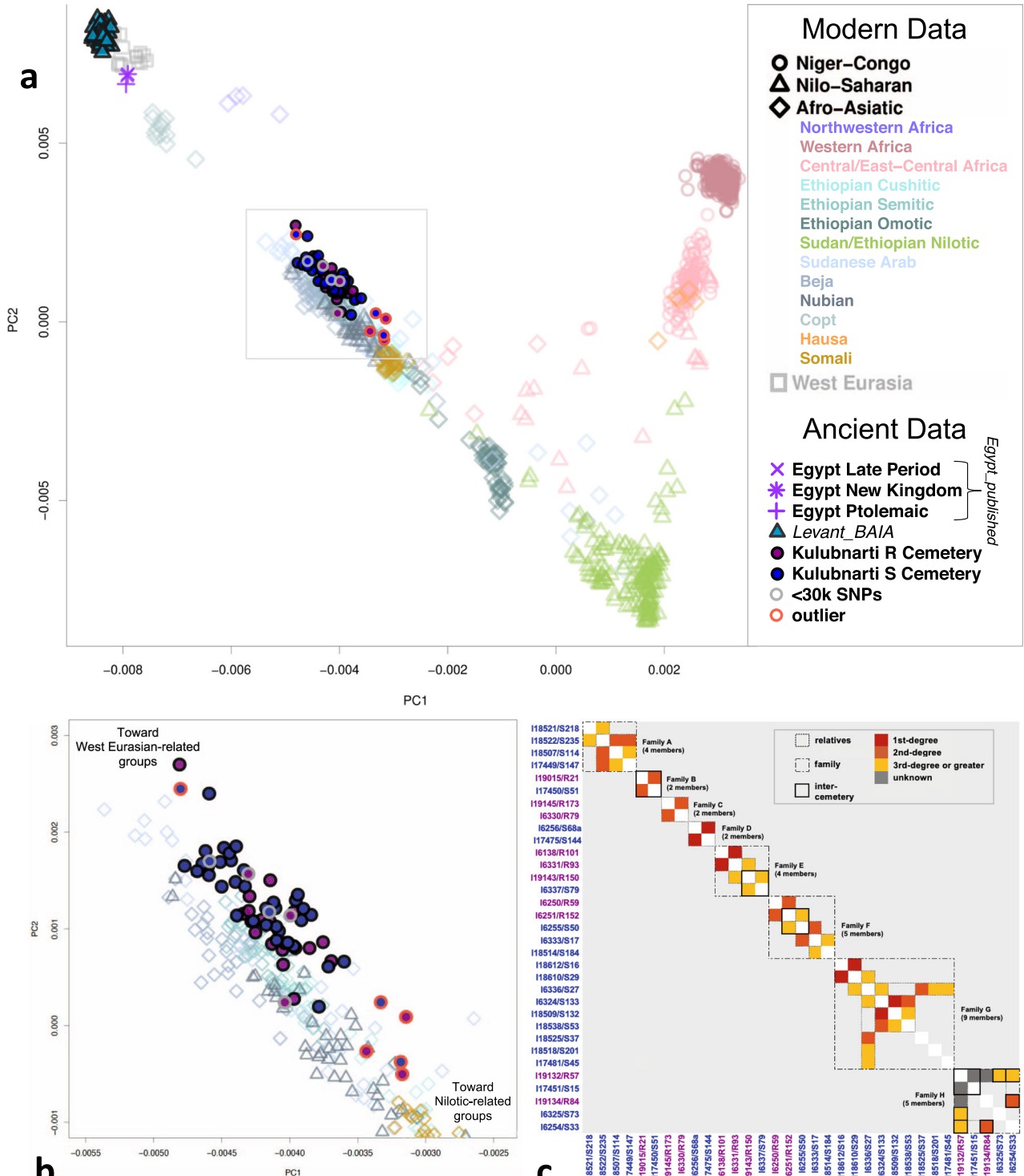

**Fig. 2 Overview of population structure at Kulubnarti. a** PCA with Kulubnarti Nubians projected onto axes computed using present-day African and West Eurasian populations. Individuals from Kulubnarti are shown as solid maroon or blue circles (representing R and S cemetery individuals, respectively), significant genetic outliers are outlined in orange, individuals with <30 K SNPs are outlined in gray. The area within the gray box is featured in (**b**). See Supplementary Data 4 for all individuals shown. **b** PCA zoomed to show the spread of the Kulubnarti Nubians along the West Eurasian–Nilo-Saharan cline. **c** Pairwise genetic relatedness estimates for all individuals from Kulubnarti. R and S cemetery individuals labeled in blue and maroon, respectively. Degree of relatedness is indicated by square color, with solid outlines denoting relatives, dotted outlines denoting families, and thick outlines denoting inter-cemetery relatives. Relationships labeled as 'unknown' have too few overlapping SNPs to determine degree of relatedness. Data are in Supplementary Table 2.

**Table 1 Distribution of relative pairs across cemeteries.**

| | Observed within-cemetery relative pairs | | Observed cross-cemetery relative pairs | Total relative pairs | Number of individuals involved in relative pairs | | Expected probability of randomly drawing cross-cemetery relatives | Expected cross-cemetery relative pairs | Observed/Expected cross-cemetery relative pairs | P-value from a binomial distribution (probability of as few cross-cemetery relative pairs as observed) |
|---|---|---|---|---|---|---|---|---|---|---|
| | R | S | | | R | S | | | | |
| 1st degree | 1 | 3 | 0 | 4 | 2 | 6 | 0.429 | 1.71 | 0% | 0.107 |
| 2nd degree | 2 | 5 | 2 | 9 | 6 | 11 | 0.485 | 4.37 | 46% | 0.105 |
| 3rd degree | 1 | 8 | 4 | 13 | 4 | 17 | 0.324 | 4.21 | 95% | 0.582 |
| 1st + 2nd degree | 3 | 8 | 2 | 13 | 8 | 16 | 0.464 | 6.03 | 33% | 0.021 |
| 1st + 2nd + 3rd degree | 4 | 16 | 6 | 26 | 10 | 22 | 0.444 | 11.53 | 52% | 0.021 |

We compute the expected probability that two randomly chosen individuals of known-degree relatedness are cross-cemetery relatives based on the number of individuals detected as part of relative pairs in each cemetery. We compute a *p*-value from a binomial distribution of detecting as many cross-cemetery relative pairs as observed or fewer.

all first- and second-degree and all first-, second-, and third-degree relative pairs (both $p = 0.021$), though this effect is not significant when relative pairs of each degree are analyzed separately (all $p > 0.105$). The rate of cross-cemetery relative pairs relative to expectation is 0% for first-degree relative pairs (0 observed compared to 1.7 expected), 46% for second-degree relative pairs (2 observed compared to 4.37 expected), and 95% for third-degree relative pairs (4 observed compared to 4.21 expected). These results show that there is some enrichment of relative pairs buried in the same cemetery versus in different cemeteries which is most evident for first- and second-degree relative pairs (Table 1), as expected if people tended to be buried with very close family members. However, the signal attenuates to a level that is indistinguishable from random at the third-degree relative level. This is consistent with a scenario where any system of social division at Kulubnarti did not prevent gene flow between plausibly stratified groups.

Despite archeological evidence suggesting low population density at Kulubnarti given its location in a region with limited food productivity and economic resources[22], we find that eight out of nine individuals with at least 400,000 SNPs covered (all from the S cemetery) have no or relatively low fractions of their genomes in runs of homozygosity (ROH) > 4 centimorgan (cM)[50] (Supplementary Note 3; Supplementary Table 3; Supplementary Fig. 2). The paucity of short ROH 4–8 cM (three of six individuals with ROH have a maximum of three ROH of this size) suggests that the mating pool of the Kulubnarti Nubians was not sufficiently limited to result in a consistently elevated rate of short ROH[51]. The finding that intermediate ROH (8–20 cM) was more common points toward Kulubnarti functioning as a small community that mostly mated among themselves but also exchanged mates with a bigger meta-population, increasing the overall size of the mating pool. We identify only one individual (I6336/S27) with ~80 cM of his genome in >20 cM blocks, as would be expected for the offspring of close genetic relatives (potentially as close as first cousins; Supplementary Table 3; Supplementary Fig. 3).

**Kulubnarti Nubians had varying proportions of Nilotic- and West Eurasian-related ancestry.** We used principal component analysis (PCA) to illustrate how the individuals from the two Kulubnarti cemeteries relate to ancient and present-day people and to each other, projecting ancient individuals onto the first two principal components (PCs) inferred from genotyped present-day African and West Eurasian populations ("Methods"; Fig. 2a, b; Supplementary Data 4). Present-day individuals are arranged along two clines that share a terminus at the bottom right of the plot near Nilo-Saharan-speaking peoples from Sudan, South Sudan, and Ethiopia. A first cline correlates to increasing proportions of West African-related ancestry, extending between Nilo-Saharan-speakers and West Africans. A second cline correlates to increasing proportions of West Eurasian-related

ancestry (in this work we use the qualifier "-related" when the ancestry we are discussing is related to that deriving from a particular geographic area but is not necessarily from that region itself), extending from Nilo-Saharan-speakers to West Eurasians. Sudanese Arab (here we use a group identifier based on ethnic and linguistic categories following the original publication that reported the data), Beja, and Nubian people from the north-eastern and central regions of Sudan, along with Afro-Asiatic-speakers from Ethiopia and Somalia, fall intermediate along this cline.

The positioning of present-day groups in PCA is consistent with Sudanese and Ethiopian people exhibiting a major axis of genetic variation based on proportion of West Eurasian-related ancestry, which is generally correlated more strongly with geography than language group[20,44,52,53]. West Eurasian-related ancestry has been present in northeastern Africa for at least 5000 years and potentially far longer[44,54–58] although we refer to this ancestry here as "West Eurasian-related" because we do not yet have ancient genetic data from an appropriate phylogenetically-adjacent reference group from Africa which is likely to have been its proximate source. The genetic structure of present-day Nubians has been influenced by a relatively recent spread of West Eurasian-related ancestry southward along the Nile and Blue Nile during the Arab conquest of the late 1st and 2nd millennia CE[20,53]. Therefore, an important question is whether a substantial proportion of West Eurasian-related ancestry was present in the Nile Valley prior to the Arab expansion, and from where such ancestry ultimately derived.

Individuals from Kulubnarti fall along a cline with Nilo-Saharan-speakers at one extreme and West Eurasian groups at the other. The Kulubnarti individuals approximately overlap present-day Sudanese Arabs, Beja, and Nubians, as well as Semitic and Cushitic-speaking Ethiopians. This suggests that the ancient Kulubnarti individuals have both West Eurasian-related ancestry and ancestry related to Nilo-Saharan-speakers (in what follows, we use the term "Nilotic" to refer to the ancestry related to the people who have lived in parts of northeastern Africa, including southern parts of Sudan, for a long period of time and who speak Nilo-Saharan languages; we emphasize that Nilo-Saharan languages are spoken over a broader region, and in this paper we do not use the term "Nilotic" to refer to Nilo-Saharan speakers outside this core region). The Kulubnarti Nubians on average are shifted slightly toward present-day West Eurasians relative to present-day Nubians, who are estimated to have ~40% West Eurasian-related ancestry[20,53]. The spread of individuals from Kulubnarti along this cline suggests individual variation in the proportion of West Eurasian- and Nilotic-related ancestry with no systematic differences in ancestry between the R or S cemetery groups.

To formally test whether the Kulubnarti Nubians were admixed, we pooled all individuals (based on qualitative similarities in PCA) and computed admixture $f_3$-statistics of the

form $f_3$(Kulubnarti; Nilotic_Test, WestEurasia_Test), where a negative statistic would indicate that, on average, the allele frequencies in the Kulubnarti population are intermediate between Nilotic_Test and WestEurasia_Test, supporting a history of admixture between people related (perhaps deeply) to these two populations. Here, we specifically use Dinka as a proxy for Nilotic-related ancestry based on evidence that groups such as the Dinka occupying the region around the White Nile show long-term genetic continuity, genetic isolation, and genetic links to ancestral East African people, and that an "unadmixed" Nubian gene pool is genetically most similar to Nilotic people[20]. We tested 32 modern and geographically and temporally-diverse ancient West Eurasian populations, also including a pool of three ancient Egyptians who had a majority proportion of West Eurasian-related ancestry ('Egypt_published', comprised of published data from two individuals from the Pre-Ptolemaic New Kingdom and Late Period and one individual from the Ptolemaic Period)[31], as WestEurasia_Test ("Methods"). Negative $f_3$-statistics ($|Z| > 7.5$) indicate that Kulubnarti Nubians were admixed between these ancestry types (Supplementary Data 5), confirming that a substantial West Eurasian-related ancestry component was present in this part of Nubia prior to the later migrations that contributed to the present-day genetic landscape. This is consistent with evidence of West Eurasian and Egyptian influence at Kulubnarti, including Christian churches as well as inscriptions in Greek and Coptic as well as Old Nubian[59,60].

Given the moderate spread of Kulubnarti Nubians along the Nilotic–West Eurasian cline when projected in PCA, we investigated whether any individuals were outliers with a significant excess of Nilotic- or West Eurasian-related ancestry relative to other individuals (here, "outlier" denotes an individual with ancestry proportions that are significantly distant from the group mean rather than an individual who is part of a distinct genetic cluster). To test this, we used the statistic $f_4$(Nilotic_Test, WestEurasia_Test; Individual, Kulubnarti_Without_Individual), again using Dinka as Nilotic_Test and using Levant_BAIA (a pool of individuals from sites in the Levant that date to the Bronze and Iron Ages, chosen because this pool comprises a sample that gives the most negative Z-score in $f_3$-statistic tests and is shown using qpAdm to be the best proxy for West Eurasian-related ancestry at Kulubnarti, described below) as WestEurasia_Test. We computed this statistic for each individual, where Kulubnarti_Without_Individual was the pool of all individuals from Kulubnarti minus the individual being tested ("Methods"; Supplementary Data 6). We consider statistics to be significant at $|Z| > 5.0$, a stringent threshold set to identify the most notable outliers amidst substantial inter-individual variation. At this threshold we identify six outlier individuals (Fig. 3), one (I18518/S201) who has significantly more West Eurasian-related ancestry ($|Z| = 7.2$), and five (I18508/S115 and I18536/S42b from the S cemetery and I6328/R201, I19135/R91, and I6252/R181 from the R cemetery) who have significantly more Nilotic-related ancestry ($|Z| > 5.1$); we removed these outliers in subsequent group-level analyses. Although I19145/R173 has the greatest proportion of West Eurasian-related ancestry estimated with qpAdm (discussed below), this individual does not pass our threshold for significant outliers ($Z = -3.9$); this may be a result of the relatively low coverage of this individual (~73K SNPs). While only six individuals were removed as genetic outliers, we note a pattern of overly-dispersed heterogeneity in terms of proportion of Nilotic- and West Eurasian-related ancestry among the Kulubnarti Nubians consistent with a scenario of relatively recent or ongoing admixture (Supplementary Fig. 4).

**Ancestry similar to that in Bronze or Iron Age Levant was likely introduced to Kulubnarti via Egypt.** To obtain insight into the relative proportions of Nilotic- and West Eurasian-related ancestry and the origin of the West Eurasian-related ancestry at Kulubnarti, we again pooled individuals, this time excluding the six genetic outliers, and applied qpAdm[29] ("Methods"; Supplementary Note 4). We selected a reference population set that allowed us to model the Kulubnarti Nubians as descended from two-way admixture between Nilotic-related and West Eurasian-related populations while also differentiating between possible sources of West Eurasian-related ancestry. We began with the "O9" reference set[34], previously used to disentangle divergent strains of ancestry in ancient West Eurasians (e.g., ref. [38]). We examined the fit of the 21 ancient populations previously used for admixture $f_3$-statistics as the West Eurasian-related source and found multiple plausible solutions for two-way admixture models ($p > 0.05$) between Dinka and Bronze or Iron Age people from the Levant (Levant_BAIA) or Anatolia (Anatolia_EBA) (Supplementary Data 7). No West Eurasian populations predating the Bronze Age fit as plausible sources, suggesting that the West Eurasian-related ancestry in the Kulubnarti Nubians is complex and itself admixed, plausibly requiring both Levantine- and/or Anatolian-related ancestry as well as a non-trivial amount of Iranian/Caucasus-related ancestry, which was spread into Anatolia and the Levant in the Chalcolithic and Early Bronze Age[34,38,40]. Complex and admixed West Eurasian-related ancestry at Kulubnarti is consistent with previous work showing that ancestry such as that found in Levant Neolithic-related populations made a critical contribution to the genetic landscape in parts of Africa several thousand years ago[54,61] and that ancestry related to the Iranian Neolithic appeared in parts of Africa after the earlier gene flow related to Levant Neolithic populations, with Iranian Neolithic ancestry identified throughout the Levant during the Bronze Age[34] and in Egypt by the Iron Age[31]. Indeed, we find that Egypt_published also fits as a West Eurasian-related source, suggesting that a similar type of West Eurasian-related ancestry was present in Egypt as well as Kulubnarti, consistent with the geographically- and archeologically-plausible scenarios that Egyptians could have been the more proximal source for the introduction of West Eurasian-related ancestry southward into ancient Nubia.

More than one two-way admixture model produced a valid fit with the O9 reference set. Therefore, we adopted a model competition approach, taking pairs of fitting models and adding the source population in one of the fitting models into the reference population set for the other fitting model and evaluating whether it continued to fit. If the model fails, this provides evidence that the source population moved to the reference population set shares genetic drift with the Kulubnarti individuals not present in the other source population, and thus the source moved to the reference set is in some sense genetically closer[34,40]. Of the three plausible sources determined with the O9 reference set (Levant_BAIA, Anatolia_EBA, and Egypt_published), all models with Levant_BAIA and Anatolia_EBA as the West Eurasian-related source fail when Egypt_published is included in the reference set ($p < 6.3\text{E}-06$), and we obtain a fit only when Egypt_published is used as the West Eurasian-related proxy ($p = 0.87$). With this model, we estimate that $60.4 \pm 0.5\%$ ancestry in the Kulubnarti Nubians is ancient Egyptian-related. However, ancient Egyptians have been shown to harbor a non-trivial amount of Dinka-related ancestry, which we re-estimate here using qpAdm to be $5.0 \pm 0.7\%$ ($p = 0.78$; Supplementary Data 7); therefore, we cannot use Egypt_published directly as a proxy source for estimating the proportion of West Eurasian-related ancestry in the Kulubnarti Nubians. Furthermore, we are interested in identifying the most precise distal source for the West Eurasian-related ancestry introduced into Nubia, and so we removed Egypt_published from our qpAdm model and performed a model competition approach again including only the two

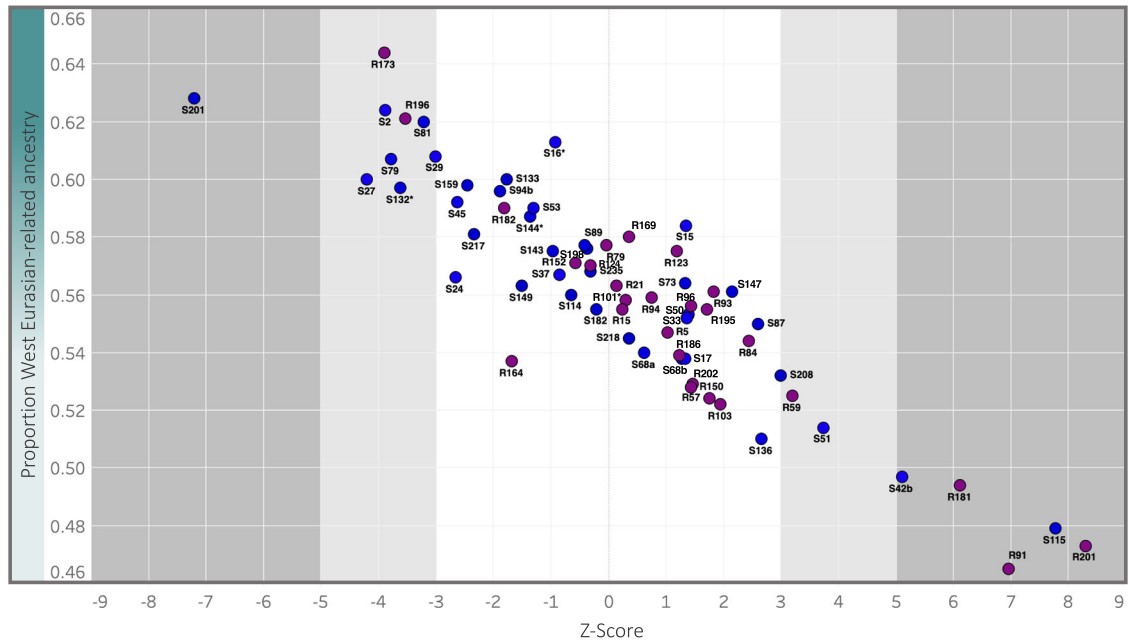

**Fig. 3 Identifying genetic outliers at Kulubnarti.** We plot the Z-score for the statistic $f_4$(*Nilotic_Test, WestEurasia_Test; Individual, Kulubnarti_Without_Individual*) for each individual at Kulubnarti (purple and blue circles represent R and S cemetery individuals, respectively) on the x-axis against the point estimate of West Eurasian-related ancestry on the y-axis (data in Supplementary Data 6). At |Z| > 5 (dark gray area), we consider individuals to be genetic outliers; individuals with |Z| > 3 (light gray area) were not considered outliers at the threshold set for this work. Asterisks (*) denote lower-coverage first-degree relatives of other individuals in the dataset.

plausible populations with origins in West Eurasia (*Levant_BAIA* and *Anatolia_EBA*). We find that the only fitting model is one using *Levant_BAIA* as a source ($p = 0.44$), and we estimate that the Kulubnarti Nubians had $57.5 \pm 0.3\%$ West Eurasian-related ancestry ultimately most like that found in Bronze or Iron Age people from the Levant, although this ancestry was likely introduced to Nubia through ancient Egyptians or a group related to them. This reflects deep biological connections among populations inhabiting the Nile Valley and further confirms the presence of West Eurasian-related ancestry in the Nile Valley prior to the later Arab migrations.

**No systematic differences in ancestry among individuals from the Kulubnarti cemeteries.** Following population-level *qpAdm* analysis, we quantified proportions of Nilotic- and West Eurasian-related ancestry for each individual using *qpAdm* ("Methods"; Supplementary Data 8; Supplementary Note 4; Supplementary Fig. 5). Non-outlier individuals from Kulubnarti had $35.6$–$49.0 \pm 1.2$–$3.3\%$ Nilotic-related ancestry, while the outlier I18518/S201 was modeled as having $37.2 \pm 1.3\%$ of such ancestry (I19145/R173 had a lower point estimate of 35.6% Nilotic-related ancestry, but was not designated as an outlier at a |Z| = 5 threshold for individual $f_4$-statistics). Outliers I19135/R91, I6328/R201, I6252/R181, I18508/S115, and I18536/S42b were modeled as having $50.3$–$53.5 \pm 1.4$–$2.6\%$ Nilotic-related ancestry, significantly more than the rest of the Kulubnarti population.

We looked for systematic differences in ancestry among individuals buried in the R and S cemeteries, inspired by the unresolved question of whether the individuals buried in the two cemeteries who experienced significant differences in morbidity and mortality broadly reflective of socioeconomic differences also had differences in genetic ancestry. When pooling individuals by cemetery (excluding outliers), we find minimal difference in average proportion of Nilotic- and West Eurasian-related ancestry between groups, with overlapping estimates of Nilotic-related ancestry inferred using *qpAdm* in *Kulubnarti_R*

($43.2 \pm 0.4\%$) and *Kulubnarti_S* ($42.3 \pm 0.4\%$) ("Methods"; Supplementary Data 7). To formally test the significance of this difference in ancestry proportions, we applied *qpWave* and obtained a value of $p = 0.25$ for a clade test between *Kulubnarti_R* and *Kulubnarti_S*, suggesting that these groups are indeed consistent with being random samples of the same population ("Methods"; Supplementary Data 9). We further confirm that *Kulubnarti_R* and *Kulubnarti_S* form a clade using the statistic $f_4$(*Dinka, WestEurasia_Test; Kulubnarti_R, Kulubnarti_S*), which is non-significant after correcting for multiple hypothesis testing (Z < 2.4) for every *WestEurasia_Test* population ("Methods"; Supplementary Data 10). Further supporting this result is an $F_{ST}$ value of 0.0013 between *Kulubnarti_R* and *Kulubnarti_S* (although we note that $F_{ST}$ is sensitive to relatedness, and there are a number of second- or third-degree relatives in our dataset) ("Methods"; Supplementary Table 4). Taken together with the multiple cross-cemetery relatives, these results indicate that the people buried in the R and S cemeteries were part of the same genetic population, an important finding in light of the observed anthropological and archeological differences between the two groups that are suggestive of socioeconomic stratification.

**Continuous waves of admixture contributed to the gene pool at Kulubnarti.** Morphological evidence has been interpreted as providing evidence for little if any extra-regional gene flow into Kulubnarti and suggests that the site was relatively isolated[62,63]; however, genetic data have higher resolution than morphological data for detecting gene flow. Inter-individual variation in proportions of Nilotic- and West Eurasian-related ancestry suggests that admixture occurred relatively recently or was indeed ongoing. To test this hypothesis, we used the *DATES* software to estimate the time since mixture using ancestry covariance patterns that can be measured in a single individual[40]. We pooled the individuals from Kulubnarti (again excluding outliers) and, using Dinka and *Levant_BAIA* as a reference pair, estimated admixture to have occurred an average of $22.2 \pm 1.4$ generations, or

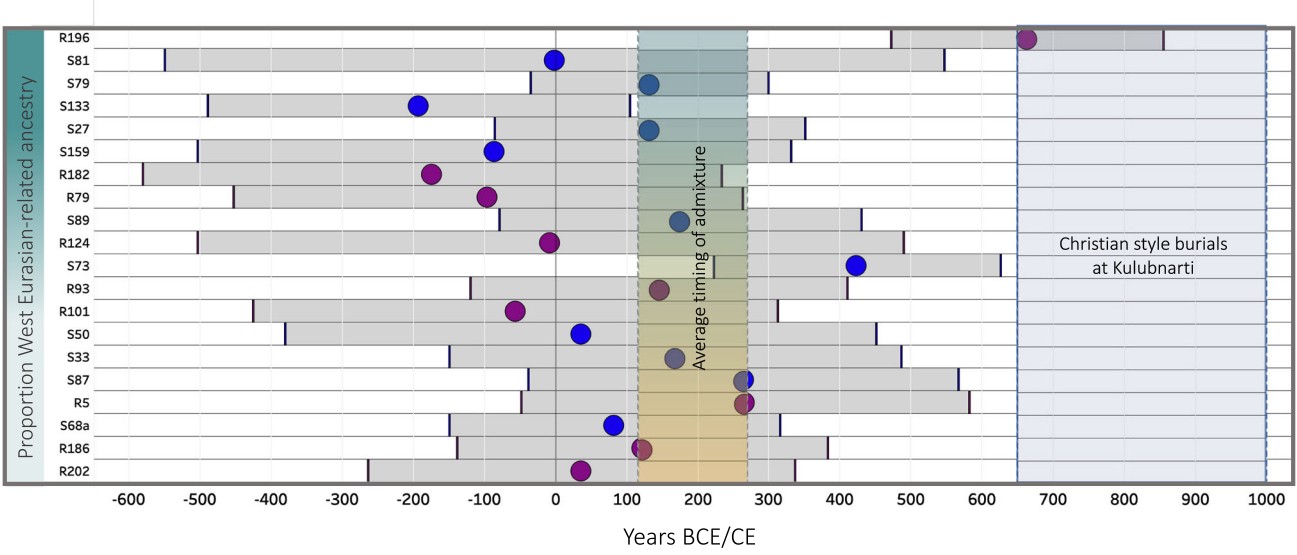

**Fig. 4 Estimated timing of admixture for 20 Kulubnarti Nubians with direct ¹⁴C dates.** Individuals are ordered from most to least amount of West Eurasian-related ancestry; maroon and blue circles represent individuals buried in the R and S cemeteries, respectively; location of the circles indicates mean estimate of admixture using *DATES* and determined with calibrated ¹⁴C date, gray horizontal bars indicate 95% CI of admixture date estimate in years BCE/CE. 95% CI of the average timing of admixture at Kulubnarti (111–265 CE) and dates of use of Kulubnarti R and S cemeteries (~650–1000 CE) shown in shaded areas. Data in Supplementary Data 11.

~620 ± 40 years (95% CI, ~700–545 years), before the studied individuals lived, assuming a generation time of 28 years[64] (Supplementary Data 11). Using 810 CE as the midpoint of the calibrated modeled age range for Kulubnarti, this places admixture occurring on average during the early-2nd to late-3rd centuries CE (95% CI), although the dates obtained with this method are based on a model of a single pulse of admixture and thus reflect an intermediate value if the true history includes multiple waves or continuous admixture, which is likely at Kulubnarti given the individual-level variance in ancestry proportions. Running *DATES* separately on the cemetery groups *Kulubnarti_R* and *Kulubnarti_S*, we found that the average admixture dates are overlapping, estimated to have occurred 21.7 ± 1.9 and 22.3 ± 1.7 generations, or ~610 ± 50 and ~625 ± 50 years (95% CI, ~715–500 years), respectively, before the studied individuals lived (Supplementary Data 11). This provides additional support for the similar population histories of the people buried in the two cemeteries.

To explore in detail whether recent or ongoing admixture contributed to the formation of the admixed gene pool at Kulubnarti, we applied *DATES* to each individual from Kulubnarti, requiring a Z-score of at least 2.8 (corresponding to a 99.5% CI) for difference from zero to be considered a valid estimate and obtaining an estimate for 32 individuals ("Methods"; Supplementary Data 11). Individual estimates of admixture dates ranged from 10.4 ± 3.5 generations (I6327/R196) to 46.2 ± 11.8 generations (I19143/R150) before the lifetime of the individuals, corresponding to admixture as recent as ~100–500 years (95% CI) and as distant as ~650–1900 years (95% CI) before the lifetime of the individual (though again, these values reflect an average if there were multiple waves or continuous admixture). Though most pairs of individuals have overlapping inferred admixture dates (95% CI), this is not true of every pair, providing evidence that the admixture did not all occur at a single time. Considering the twenty individuals who also have ¹⁴C dates as well as *DATES* estimates allows us additional insight into the timing of admixture. Here, using calibrated ¹⁴C dates, we observe point estimates ranging from ~200 BCE (95% CI, ~490 BCE–100 CE) to ~660 CE (95% CI, 470–850 CE) (Fig. 4). We computed

Z-scores for the difference between all possible pairs (Supplementary Table 5) and found significant variation in inferred admixture dates after Bonferroni correction for significance based on the number of hypotheses tested (|Z| > 3.65), confirming that there is significant variation in average admixture date estimates and suggesting that waves of admixture extending on the order of a millennium contributed to the formation of the Kulubnarti gene pool.

**West Eurasian-related ancestry in Kulubnarti disproportionately derived from female ancestors.** Previous morphological analysis has found no evidence of sex-specific patterns of mobility at Kulubnarti[62], while analysis of Y chromosome and mitochondrial DNA (mtDNA) are unlikely to paint a clear picture of sex-biased ancestry because many haplogroups that are common in West Eurasia are also found at high frequencies in parts of northeastern Africa; however, genome-wide data provide a potentially more powerful way to investigate sex-biased ancestry. To test for evidence that West Eurasian-related ancestry may have been introduced into Kulubnarti in a sex-biased way, we analyzed male and female demographic histories separately. Because females carry two-thirds of the X chromosomes in a population but only half of the autosomes, the X chromosome can be used to detect a signal of asymmetrical admixture between males and females[65]. We therefore used our *qpAdm* model to compute ancestry proportions on the autosomes and the X chromosome (as in ref. [32]) of the Kulubnarti Nubians (excluding genetic outliers; "Methods"). We found that West Eurasian-related ancestry (modeled using *Levant_BAIA* as a proxy) accounts for 57.5 ± 0.3% of the autosomes but 64.4 ± 1.8% of the X chromosomes in the Kulubnarti Nubians (Z = 3.8; Table 2), revealing that West Eurasian-related ancestry at Kulubnarti disproportionately derived from female ancestors. Our point estimate of the proportion of West Eurasian-related ancestry deriving from females is 68%, with a 95% CI of 59–77% ("Methods").

Examining uniparentally inherited parts of the genome ("Methods"; Supplementary Data 12; Supplementary Note 5;

**Table 2 *qpAdm* models for sex-biased admixture estimated using the autosomes and the X chromosome.**

| Population | Autosomes | | | | X chromosome | | | | Z-score for X-autosome difference |
|---|---|---|---|---|---|---|---|---|---|
| | Nilotic-related | West Eurasian-related | Std. Err. | *P*-value | Nilotic-related | West Eurasian-related | Std. Err. | *P*-value | |
| Kulubnarti | 42.5% | 57.5% | 0.3% | 0.435 | 35.60% | 64.4% | 1.8% | 0.988 | 3.78 |
| Kulubnarti_S | 43.2% | 56.8% | 0.4% | 0.289 | 36.20% | 63.8% | 1.7% | 0.970 | 4.01 |
| Kulubnarti_R | 42.3% | 57.7% | 0.4% | 0.382 | 33.70% | 66.3% | 2.9% | 0.638 | 2.94 |

'O9' + *Anatolia_EBA* used as the reference set ("Methods"). Ancestry proportions and standard error computed by *qpAdm*; Z-score represents difference in West Eurasian-related ancestry between the autosomes and X chromosome; the Z-score is positive if there is more West Eurasian-related ancestry on the X chromosome (i.e., female-biased ancestry); formula for Z-score calculation in "Methods". We use Dinka and *Levant_BAIA* as proxy sources for Nilotic-related and West Eurasian-related ancestry, respectively.

Supplementary Fig. 6), we find that 35 out of 63 individuals from both cemeteries who were not first-degree relatives sharing a maternal lineage belong to 11 mitochondrial DNA (mtDNA) haplogroups that are presently distributed predominantly in West Eurasia, although the presence of such lineages for thousands of years in northeastern Africa as well has been established by previous work[31,66,67]. The observation of 35 individuals carrying mtDNA haplogroups that are most common in West Eurasia is what would be expected for anywhere between 43 and 68% of maternal ancestry at Kulubnarti coming from West Eurasian ancestors via northeastern Africa (based on evaluating whether each proportion in this range included 35 West Eurasian mitochondrial haplogroups within its 95% central CI), which overlaps the 59–77% estimate of West Eurasian-related ancestry deriving from females made by comparing ancestry proportions on the autosomes and X chromosome ("Methods").

Thirteen individuals from both cemeteries belong to H2a, a European-centered mtDNA haplogroup not previously found in ancient contexts in Africa to our knowledge. Upon closer examination, the presence of three additional mutations not typically found in members of this haplogroup suggests that they are likely part of a previously undocumented branch of H2a. Ten individuals from both cemeteries belong to mtDNA haplogroup U5b2b5, though they also exhibit three additional mutations not typically found in members of this haplogroup. One of these mutations was detected in a 4000-year-old mummy from Deir el-Bersha, Egypt also assigned to this haplogroup[67], raising the possibility that the presence of U5b2b5 at Kulubnarti reflects deep connections with Egypt; other mtDNA haplogroups, including J2a2e, R0a1, T1a7, U1a1, and U3b are also found both at Kulubnarti and in ancient Egyptians[31]. U1a1, U3b, and N1b1a2 have also been identified in Bronze Age individuals from Israel and Jordan[42], so the presence of these lineages also at Kulubnarti is consistent with the genome-wide data. Previously published ancient Egyptian data in ref. [31] includes only one individual belonging to an African-originating L mtDNA haplogroup, suggesting that female-specific African ancestry may have had a limited impact as far north as Egypt as late as the Roman Period[31]. In contrast, 28 individuals from Kulubnarti belong to seven different L lineages, the most common being the eastern African sub-clade L2a1d1, supporting a deep matrilineal connection to this region[68]. Four individuals from Kulubnarti belong to L5a1b, a lineage of the rare L5a haplogroup centered in East Africa. mtDNA haplogroup L5 has been observed only at low frequency in East and Central Africa and also in Egypt[69–71]; the L5a1b lineage has previously been identified in a Pastoral Neolithic individual from Hyrax Hill in Kenya dating to ~2300 years BP[58].

We called Y chromosome haplogroups for 30 males from Kulubnarti, 28 of whom are not first-degree relatives that share a patriline ("Methods"; Supplementary Data 13; Supplementary Fig. 7). Seventeen males (including two pairs of relatives with a

shared patriline) belong to haplogroups on the E1b1b1 (E-M215) branch that likely originated in northeast Africa ~25 kya[72] and is commonly found in present-day Afro-Asiatic speaking groups[73]. In this subset of males, E1b1b1a1a1c (E-Y125054) was the most common haplogroup, called for a father–son pair from the S cemetery as well as two unrelated individuals from the R cemetery. Of the 15 unrelated males assigned to branches of E1b1b1, 10 were buried in the R cemetery. While 5 males from the S cemetery belonged to haplogroups on the E1b1b1 branch and another belonged to E2a (E-M41), nine belonged to Y haplogroups with likely West Eurasian origins—albeit also with distributions that include northeastern Africa—compared to only three from the R cemetery. While males from the R cemetery are more likely to belong to haplogroups on the Y chromosome E branch, the most represented Y lineage in Africa[74], the difference is not significant ($P = 0.11$, "Methods"), and so we view this as likely to be a statistical fluctuation and do not take this as evidence of heterogeneity among males from each cemetery.

**Insight into present-day Nubian people.** We were interested if ancient DNA data from Kulubnarti could provide new insights into the processes that shaped the genomes of genotyped present-day Mahas, Danagla, and Halfawieen Nubian populations (reported in ref. [20]). First, we show using *qpWave* that none of these three present-day Nubian populations form a clade with the Kulubnarti Nubians and are therefore not their direct descendants without additional admixture ($p < 1.5E-10$ for clade tests) ("Methods"; Supplementary Data 9). Using *DATES* and the same reference pairs as previously mentioned, we re-estimate that admixture occurred on average $33.9 \pm 2.5$ generations ago for the Mahas (95% CI, 889–1210 CE), $36.6 \pm 2.2$ generations for the Danagla (95% CI, 855–1095 CE) and $24.2 \pm 3.2$ generations for the Halfawieen (95% CI, 1148–1498 CE) ("Methods"; Supplementary Data 14). These dates are consistent with those estimated in Hollfelder et al. (2017), supporting their conclusion that migrations that occurred with the Arab conquest beginning in the 7th century[1] left a detectable signature on the genomes of present-day Nubian peoples. We tested if the same *qpAdm* model that explained ancestry in the Kulubnarti Nubians could also be applied to these present-day Nubian people, but found that the model did not fit any present-day Nubian group tested ($p < 2.3E-07$); we were also not able to fit Kulubnarti as a source in a two-way admixture model for any present-day Nubian group ($p < 2.1E-06$) ("Methods"; Supplementary Data 7). We can therefore assume that the admixture events that shaped the genomes of these present-day Nubians also introduced different types of ancestry into the gene pool of these people. Thus, despite a superficial resemblance on our PCA, the present-day Nubian populations for which we have genotype data are not descended from a population related to the earlier Kulubnarti Nubians without additional admixture following the Christian Period. Studies of genetic patterns in modern Nubians—both from the

perspective of genome-wide data[20] and from the perspective of uniparental markers like mtDNA[75]—are thus important for informing on present-day populations, but our results show that these results cannot be simply extrapolated back to ancient Nubians. Instead, this requires ancient DNA data, such as we report on here.

## Discussion

Driven by questions inspired by archeology and bioarcheology, our analysis provides new insight into the ancestry of Christian Period people from Kulubnarti and into the genetic relationships among individuals buried in two cemeteries with significant differences in morbidity and mortality suggestive of social stratification.

First, we find that all individuals from Kulubnarti were admixed with varying amounts of Nilotic- and West Eurasian-related ancestry. A high proportion of West Eurasian-related ancestry, ultimately deriving from an ancestry pool like that found in the Bronze and Iron Age Levant, is consistent with archeological evidence showing cultural influence of ultimate West Eurasian origin at Kulubnarti; specifically, Christian churches, Christian-style burials oriented east-to-west and lacking grave goods, and inscriptions in Greek and Coptic as well as Old Nubian demonstrate a transition to new practices following the introduction of Christianity into Nubia[76]. Leveraging previously published genome-wide data from three ancient Egyptians, we show that the West Eurasian-related ancestry detected at Kulubnarti was plausibly introduced via people from Egypt who harbored a majority of Western Eurasian-related ancestry and a minor proportion of Nilotic-related ancestry. The introduction of West Eurasian-related ancestry through Egypt is consistent with archeological evidence of connections between Egypt and the Levant established by the first half of the 4th millennium BCE[77,78] and between Egypt and Nubia ongoing since at least the second half of the 3rd millennium BCE[1–4,79,80]. Archeological and strontium isotope studies have identified Egyptian occupation as far as southern Upper Nubia[4,79,81,82] and have uncovered well-established cultural and material links between Nubia and the Ptolemaic and Roman Egyptian and Hellenistic worlds existing alongside indigenous cultural traditions rooted in Sudanic Africa[79,83]. Studies of skeletal morphology[60,63] and genetic studies of present-day populations[52] suggest long-term interactions between Egypt and Nubia involving gene flow. It has even been suggested that the Kulubnarti Nubians could have migrated into the *Batn el Hajar* from the north[21]. We now provide further support for biological connectivity between Kulubnarti and more northern parts of the Nile Valley using ancient DNA.

Second, we address a long-standing question about the genetic relationship among individuals buried in the R and S cemeteries at Kulubnarti. In line with historic[1], archeological[21,59], bioarcheological[8,9,62], and isotopic[84] evidence suggesting a close biological and cultural relationship among all individuals from Kulubnarti, we find no genetic evidence that people buried in the S cemetery were genetically different from people buried in the R cemetery, thus providing no support for hypotheses that they were foreign slaves, ethnic immigrants, or refugees with a distinct geographic origin or population history[21]. We identify seven pairs of cross-cemetery relatives as close as second-degree, and find that while very close family members (first- and second-degree) were likely to be buried in the same cemetery, the signal is indistinguishable from random at the level of third-degree relatives (Table 1). Here, ancient DNA provides a new line of evidence supporting the hypothesis that the burial of people in two cemeteries at Kulubnarti was not strongly rooted in genetic differences. Instead, this burial pattern may reflect social or

socioeconomic differences that are not yet fully known, or may be a cultural practice, such as the burial of unbaptized individuals or those suffering from particular illnesses, apart from the rest of the population.

Third, we show that the admixture events that contributed to the gene pool at Kulubnarti spanned roughly a millennium, with ongoing and relatively recent admixture contributing to substantial inter-individual variance in ancestry proportions. The rise (~300 BCE) and collapse (~350 CE) of the Meroitic Kingdom in Nubia provides a possible historical context for admixture between Egyptian peoples carrying West Eurasian-related ancestry and local Nubians (who may also have already had some amount of West Eurasian-related ancestry by this time, a process that would be further clarified by additional ancient DNA analysis of older individuals from Nubia). Following a period of Nubian rule of Egypt from a seat of power at Napata (around the Fourth Cataract) terminated by invading Assyrians, the Nubian kingdom of Meroë was established. While the emergence of the Meroitic Kingdom remains poorly understood, a characteristically Nubian culture developed, albeit still exhibiting strong cultural connections with Egypt and the Greco-Roman world[79,80,59,85]. During this time, areas north of the Third Cataract (including the *Batn el Hajar*) were sparsely settled and were responsible for maintaining trade and communications with Egypt, placing them in direct contact with both Egyptian powers to the north and the Meroitic kingdom to the south[85]. As such, a plausible ancestry source would be admixed groups of people living in Upper Egypt or Lower Nubia, who moved southward to Kulubnarti while continuing to exchange genes with surrounding populations. This possibility should be investigated through further ancient DNA analysis of Egyptians and Nubians predating the Christian Period.

Fourth, we find that our data are consistent with a greater amount of female mobility (and possibly exogamy). We show that West Eurasian-related ancestry at Kulubnarti was disproportionately associated with female ancestors, highlighting the importance of female mobility in this region. In line with this, although the population size at Kulubnarti is assumed to be small based on the site's location in the *Batn el Hajar*, analysis of ROH points to limited population-level relatedness and a relatively large mating pool at Kulubnarti, implying connections with a broader population. It is possible that these connections are primarily female-mediated, and that Kulubnarti was a patrilineal and patrilocal society that followed a system of patrilineal primogeniture. While this is speculative, additional ancient DNA data interpreted within an archeological framework from other parts of Nubia will benefit this discussion in the future.

Finally, in support of previous findings that present-day Nubians were influenced genetically by additional waves of admixture that post-date the Christian Period, we find no evidence that they descended directly from the Kulubnarti Nubians. Instead, interactions involving gene flow continued following the Christian Period, with estimated dates of admixture suggesting that the Arab conquest of Egypt and Sudan influenced not only the cultural landscape, but also the genetic landscape of this region. Taken together, our results reveal a dynamic population history in Nubia that began thousands of years ago and continues into the present.

## Methods

**Ancient DNA analysis.** Petrous bones were selected from the Kulubnarti osteological collection, curated at the University of Colorado at Boulder (USA) at the time of sample collection (May 2015). In ancient DNA-dedicated cleanroom facilities at University College Dublin (UCD; Ireland) or Harvard Medical School (HMS; Boston, Massachusetts, USA), we processed petrous bones from 111 individuals (see Supplementary Data 1 for all individuals analyzed and Supplementary Data 2 for the location where bone processing took place). We generated powder

following a technique that uses a dental sandblaster to systematically locate, isolate, and clean the cochlea[86]. During all bone processing and subsequent wet-lab work, we implemented appropriate criteria to prevent contamination with present-day DNA or cross-contamination with other ancient samples by working only in dedicated ancient DNA cleanroom facilities, requiring proper cleanroom attire, and implementing chemical cleaning and use of ultraviolet (UV) irradiation to minimize contamination[87]. We extracted DNA in a dedicated ancient DNA laboratory at HMS following published protocols[88–90]. We prepared dual-barcoded double-stranded[91] or dual-indexed single-stranded[92,93] sequencing libraries from all extracts (see Supplementary Data 2 for library type); for two samples (I17475/S144 and I17477/S24), we increased coverage by preparing two libraries for each sample and pooling the data. We treated all sequencing libraries with uracil-DNA glycosylase (UDG) to reduce the rate of characteristic ancient DNA damage[91]. Double-stranded libraries were treated in a modified partial UDG preparation ('half'), leaving a reduced damage signal at both ends (5′ C-to-T, 3′ G-to-A). Single-stranded libraries were treated with *E. coli* UDG (USER from NEB) that inefficiently cuts the 5′ Uracil and does not cut the 3′ Uracil.

To generate genome-wide SNP capture data for the individuals from Kulubnarti, we used an in-solution target hybridization to enrich for sequences that overlap the mitochondrial genome and 1,233,013 genome-wide SNPs ('1240k SNPs')[27–30]. We added two 7-base pair indexing barcodes to the adapters of each double-stranded library (in contrast, single-stranded libraries are already indexed from the library preparation process), and sequenced these libraries using either an Illumina NextSeq500 instrument with 2 × 76 cycles or an Illumina HiSeqX10 instrument with 2 × 101 cycles and reading the indices with 2 × 7 cycles (in the case of double-stranded libraries) or 2 × 8 cycles (in the case of single-stranded libraries). See Supplementary Note 2 for additional details.

After sequencing, we merged paired-end sequences, retaining reads that exhibited no more than one mismatch between the forward and reverse base if base quality was ≥20, or three mismatches if base quality was <20. We used a custom toolkit (https://github.com/DreichLab/ADNA-Tools) to merge sequences and to trim adapters and barcodes from the merged read. We mapped merged sequences to the reconstructed human mtDNA consensus sequence (*RSRS*; Behar et al. 2012) and the human reference genome (version hg19) using the samse command in BWA v.0.7.15-r1140[94] with the parameters -n 0.01, -o 2, and -l 16500. Duplicate molecules (those exhibiting the same mapped start and end position as well as the same strand orientation) were removed following alignment using the Broad Institute's Picard MarkDuplicates tool (v2.17.10; http://broadinstitute.github.io/picard/). To reduce damage-induced errors, we trimmed two terminal bases from all UDG-half libraries.

We evaluated the authenticity of the generated DNA data, retaining 66 individuals with at least 20,000 SNP targets hit, a minimum of 3% C-to-T substitutions at the end of the sequenced fragments[91], point estimates of matching to the mitochondrial DNA consensus sequence made using *contamMix* v.1.0–12 of >95%[28], and point estimates of X chromosome contamination (for males with at least 200 SNPs covered at least twice) below 3%[95]. Three out of 66 individuals were retained despite narrowly missing one of the set thresholds for authenticity. Two individuals (I6324/S133 and I6337/S7) were below our ~3% threshold for C-to-T substitutions at the end of the sequenced fragments, exhibiting rates of 2.3% and 2.8%, respectively (the average rate of C-to-T substitutions in these individuals was ~5.4%); however, these samples passed all other authenticity metrics and were therefore maintained. A single individual (I6340/R169) was above our threshold of 3% for X chromosome contamination (3.3%), but passed all other authenticity metrics and was also maintained.

For these 66 individuals, we created 'pseudo-haploid' SNP calls by randomly sampling an overlapping read with minimum mapping quality ≥10 and base quality ≥20. Individuals with fewer than 20,000 SNPs covered were automatically excluded from analysis. Library information for all individuals retained in this analysis is is in Supplementary Data 2. One individual from each pair of four first-degree relatives in the dataset was excluded from population genetic analysis; in all cases, we retained the higher coverage individual (Supplementary Data 1).

**Dataset assembly.** We merged our genome-wide SNP data from the 66 ancient individuals from Kulubnarti that passed quality control into a publicly available dataset that included genotypes of ancient African and Near Eastern individuals (see Supplementary Data 3 for relevant citations, a list of individuals, and the abbreviations used for these individuals throughout this analysis) restricted to a canonical set of 1,233,013 SNPs[27–30], as well as whole-genome sequence (WGS) data from present-day African and Near Eastern groups[43,45–49]. This '1240k SNP' dataset was used in this work for *f*-statistics, *qpAdm/qpWave*, relatedness, and shared genomic segments, and *DATES* analyses of ancient Nubians, unless otherwise noted. We created a second dataset ('1240k_Hollfelder'), merging published array-genotyped individuals from Sudan and South Sudan[20] with our initial '1240k SNP' dataset, yielding a total of 580,295 overlapping autosomal SNPs. We used this dataset for *DATES* analysis of present-day Nubians or when a present-day Nubian population was included in an analysis. We created a third dataset ('1240k_HollfelderPagani'), merging the '1240k_Hollfelder' dataset with an additional set of array-genotyped individuals from Ethiopia, Somalia, and South Sudan[44], yielding a total of 396,506 overlapping autosomal SNPs. We used this dataset for PCA. These three different datasets were used in different analyses to maximize coverage. All genome-wide analyses were performed on autosomal data.

In all datasets, we pooled previously published ancient individuals overlapping geographically or temporally, who were previously shown to be relatively genetically homogenous, into a single population for the purpose of increasing statistical resolution. All individuals comprising merged populations listed in Supplementary Data 3, with merged groups unique to this study are as follows:

'*Egypt_published*': comprising published data from two individuals from the Pre-Ptolemaic New Kingdom and Late Period and one individual from the Ptolemaic Period[31];

'*Levant_BAIA*': comprising published data from individuals from the Bronze Age and Iron Age Levant (Israel and Jordan) who were not differentially related (|Z| < 3.0) to the Kulubnarti Nubians when tested pairwise in their site-level genetic grouping with the statistic $f_4(Mbuti, Kulubnarti; Test1, Test2)$. This group comprises 47 individuals, published in ref. [42] unless otherwise noted: 3 Bronze Age individuals from 'Ain Ghazal[34], 3 Bronze Age individuals from Tel Hazor, 20 individuals from the Intermediate Bronze Age to the Early Iron Age from Tel Meggido (genetic outliers, low coverage individuals, and first-degree relatives excluded from the dataset in ref. [42]), 19 Bronze Age individuals from the Baq'ah (excluding first-degree relatives from the dataset), and 2 Bronze Age individuals from Tel Shaddud[37].

**Uniparental haplogroups.** We determined mtDNA haplogroups using the mitochondrial capture bam files, aligning data to the *RSRS*[96], restricting to reads with MAPQ ≥ 30 and base quality ≥20 and trimming two base pairs to remove deamination artifacts. We first constructed a consensus sequence with samtools v.1.3.1. and bcftools v.1.10.2[97] using a majority rule and then called haplogroups with Haplogrep Classify v.2.2.8[98] and the –rsrs flag. Haplogroup calls and reported mutations are based on PhyloTree version 17[99] and are found in Supplementary Data 12, with details about mtDNA haplogroup calling in Supplementary Note 5. Distributions of mtDNA haplogroup calls for 63 individuals who were not first-degree relatives sharing a maternal lineage divided by cemetery of burial and grouped by most likely geographic region of origin and primary distribution are in Supplementary Fig. 6.

We determined Y chromosome haplogroups using both targeted SNPs as well as off-target sequences that aligned to the Y chromosome based on comparisons to the Y chromosome phylogenetic tree from Yfull version 8.09 (https://www.yfull.com/). We provide two notations for Y chromosome haplogroups in Supplementary Data 13. The first notation uses a label based on the terminal mutation, while the second describes the associated branch of the Y chromosome tree based on the nomenclature of the International Society of Genetic Genealogy (http://www.isogg.org) version 15.73 (July 2020). After noting that there were a larger number of males in the R cemetery than in the S cemetery with African-associated Y haplogroups, we computed a chi-square statistic to determine if this observation was statistically significant. After Yates correction, we found that $X^2$ (1, $N = 28$) = 2.5, $p = 0.11$ and concluded that the association was not statistically significant. Distributions of Y chromosome haplogroup calls for 28 males divided by cemetery of burial who were not first-degree relatives and by most likely geographic region of origin and primary distribution are in Supplementary Fig. 7.

**Principal component analysis.** We performed principal component analysis (PCA) with smartpca[100] v.18162, using the option 'lsqproject: YES', to project ancient individuals onto the eigenvectors computed from modern individuals. The projecting of each ancient sample onto patterns of genetic variation learned from modern individuals enables us to use data from a large fraction of SNPs covered in each individual, thereby maximizing the information about ancestry that would be lost if we were restricted to a potentially smaller number of SNPs for which there is intersecting data across lower-coverage ancient individuals. We also used the option 'newshrink: YES' to remap the points for the samples used to generate the PCA onto the positions where they would be expected to fall if they had been projected, thereby allowing the appropriate co-visualization of projected and non-projected individuals. We projected 116 ancient individuals (66 newly reported and 50 previously published) onto the first two principal components computed using 811 present-day individuals from 52 populations (Fig. 2a, b). See Supplementary Data 4 for all individuals included in PCA (and whether they were used to compute axes) and values of PCs 1 and 2 for all individuals.

**Genetic relatedness.** We assessed genetic relatedness for every pair of individuals from Kulubnarti following the method described in ref. [41] and present results for first-, second-, and third-/fourth-degree relatives in Fig. 2c and in Supplementary Table 2, with additional details about the method used to assess relatedness in Supplementary Note 3. In our dataset of 66 individuals, we identify 33 individuals sharing 28 unique pairwise relationships up to the third/fourth degree. Four pairs of individuals were identified as first-degree relatives and 22 pairs of individuals were identified as second- or third/fourth-degree relatives; two relative pairs had an unknown relationship due to low overlapping SNP coverage.

**Analysis of consanguineous genomic segments.** We identified Runs of Homozygosity (ROH) within the Kulubnarti Nubian individuals using the Python package hapROH (https://test.pypi.org/project/hapROH/). Following a previously

described method[50], we used 5008 global haplotypes from the 1000 Genomes project haplotype panel[47] as the reference panel. Following the recommendations for pseudo-haploid data, we analyzed ancient individuals with a minimum coverage of 400,000 SNPs ($n = 9$) and called ROH longer than 4 centiMorgan (cM). We used the default parameters of *hapROH*, which are optimized for ancient data genotyped at a similar number of sites. For each individual, we grouped the inferred ROH into length categories >4 cM and >20 cM. We report the total sum of ROH in these length bins for each individual with sufficient coverage in Supplementary Table 3. See Supplementary Fig. 2 for visualization of ROH in each individual, illustration of ROH blocks >4 cM for individual I6336/S27, and Supplementary Note 3 for an overview of consanguinity analysis.

***f*-statistics**. We computed $f_4$-statistics using the *qpDstat* program from ADMIXTOOLS[101] with default parameters and 'f4 mode: YES'. We computed $f_3$-statistics using the *qp3Pop* program in ADMIXTOOLS using default parameters and 'inbreed: YES'.

**qpAdm**. We used *qpAdm*[29] v.1210 from ADMIXTOOLS with the option 'allsnps: NO' to identify the most likely sources of ancestry and proportions of ancestry in the Kulubnarti Nubians as well as for present-day Nubian groups. *qpAdm* uses $f_4$-statistics to detect shared drift between the target population and the possible admixing source populations, relative to a set of differentially related outgroup populations, referred to as the reference population set (see refs. [38,102]). For models that are consistent with the data ($p > 0.05$), *qpAdm* estimates proportions of admixture for the target population from the specified source populations without requiring an explicit model for how the reference populations are related. See Supplementary Note 4 for more details about *qpAdm* analysis.

**Estimating sex-biased gene flow using nuclear data**. To analyze potential sex bias in West Eurasian-related admixture, we used *qpAdm* and our single fitting admixture model (described above) to estimate admixture on the autosomes (using default parameters) and on the X chromosome (using the option 'chrom: 23') (as in ref. [32]). We computed Z-scores for the difference in ancestry proportions on the autosomes and the X chromosome as $Z = \frac{p_A - p_X}{\sqrt{\sigma_A^2 + \sigma_X^2}}$, where $p_A$ is the proportion of West Eurasian-related admixture on the autosomes, $p_X$ is the proportion of West Eurasian-related admixture on the X chromosome and $\sigma_A$ and $\sigma_X$ are the corresponding jackknife standard errors. A positive Z-score thereby represents more Nilotic-related ancestry on the autosomes (i.e., male-biased ancestry) and West Eurasian-related ancestry on the X chromosome (i.e., female-biased ancestry). See Table 2 for results.

We also estimate the proportion of West Eurasian-related ancestry driving from females. Considering $X$ to be the proportion of West Eurasian-related ancestry on the X chromosome, $A$ to be the proportion of West Eurasian-related ancestry on the autosomes, $F$ to be the proportion of female-derived West Eurasian-related ancestry, and $M$ to be the proportion of male-derived West Eurasian-related ancestry, then a simple model of admixture would be $X = \frac{2}{3}F + \frac{1}{3}M$ and $A = \frac{1}{2}F + \frac{1}{2}M$. As such, $F = 3X - 2A$ and $M = 4A - 3X$, and the proportion of West Eurasian-related ancestry coming from females is therefore $P = \frac{F}{F+M} = \frac{3X - 2A}{2A}$, giving a point estimate of 68% (95% CI 59–77%).

**qpWave**. We used *qpWave* from ADMIXTOOLS[101] with the option 'allsnps: NO' to estimate the minimum number of ancestry sources needed to form a group of test populations relative to a set of differentially related reference populations[103,104]. If the test group contains two populations, the *qpWave* methodology evaluates if they can be modeled as descending from the same sources (i.e., if they form a clade relative to the reference population set). First, we applied *qpWave* to investigate the minimum number of streams of ancestry required to model *Kulubnarti_R* and *Kulubnarti_S* relative to the O9 + *Anatolia_EBA* reference set, accepting a clade at a threshold of $p > 0.05$. Next, we used *qpWave* to assess if any of the three present-day Nubian populations could be modeled as descending from the same sources as the Kulubnarti Nubians relative to the same reference set, again interpreting cladality at $p > 0.05$. See Supplementary Data 9 for results.

**Estimation of F$_{ST}$ coefficients**. To measure genetic differentiation between *Kulubnarti_R* and *Kulubnarti_S*, we estimated $F_{ST}$ and its standard error via block-jackknife using smartpca[100] v.18162 and the options 'fstonly: YES' and 'inbreed: YES'. We removed the individual with lower coverage from each of four pairs of first-degree relatives and did not include six ancestry outliers (see main text) for this analysis. Data in Supplementary Table 4.

**Dating admixture**. We used the method *DATES* (Distribution of Ancestry Tracts of Evolutionary Signals)[40] v.3520 to estimate the average date of admixture for the pooled group of Kulubnarti Nubians (excluding one from each pair of first-degree relatives and genetic outliers), *Kulubnarti_R* and *Kulubnarti_S* (excluding one from each pair of first-degree relatives and genetic outliers), and each individual separately (including each individual from pairs of first-degree relatives and genetic

outliers). The *DATES* method measures the decay of ancestry covariance to infer the time since mixture and estimates jackknife standard errors. We used Dinka and *Levant_BAIA* as a reference pair, prioritizing high-quality data by restricting to individuals with >400,000 SNPs for increased resolution (as recommended) and using the parameters 'binsize: .001' and 'maxdis: 1.0' to ensure that we detected even relatively long LD blocks indicating more recent admixture. Results are presented in Supplementary Data 11.

We also used the *DATES* method to estimate dates of admixture for three present-day Nubian groups[20], first using Dinka and *Levant_BAIA* as a reference pair and then also making estimates with Nuer and TSI as a reference pair in order for more precise comparability with ref. [20] (Supplementary Data 11). The estimates made using both sets of reference populations were similar. The greatest difference seen when using these two sets of reference pairs was for Halfawieen, where the use of Nuer and TSI as a reference pair resulted in an estimate ~1.5 generations (~40 years) more recent than the estimate made with the Dinka and *Levant_BAIA* reference pair.

**Radiocarbon dating**. We performed radiocarbon ($^{14}$C) dating and Bayesian chronological modeling at the Center for Applied Isotope Studies (CAIS), University of Georgia (USA) for 29 individuals from Kulubnarti that yielded genome-wide data. See Supplementary Data 2 for results and Supplementary Note 2 for details regarding radiocarbon dating and modeling. Modeled dates in Supplementary Fig. 1.

**Reporting summary**. Further information on research design is available in the Nature Research Reporting Summary linked to this article.

## Data availability
The aligned sequences are available through the European Nucleotide Archive under accession number PRJEB42975 (https://www.ebi.ac.uk/ena/browser/view/PRJEB42975). Genotype datasets used for analysis are available at https://reich.hms.harvard.edu/datasets. Previously published ancient data used in this study are available under accession numbers PRJEB37057, PRJEB24794, PRJEB8448, PRJEB11450, PRJEB22652, PRJEB22629, PRJEB32466, PRJEB27215, PRJEB6272, PRJEB14455, PRJEB20914, PRJEB30874, and ERP017224 and at https://reich.hms.harvard.edu/datasets. Genotype data from present-day individuals, publicly available, were accessed as indicated in their corresponding original publications[20,43–49]. The hg19 reference genome is publicly available under GenBank assembly accession GCA_000001405.1.

## Code availability
Custom code used in this manuscript is available at https://github.com/DreichLab/ADNA-Tools.

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

## Acknowledgements

We acknowledge the ancient and present-day people whose genetic data we analyzed in this work. We are grateful to Dr. George Armelagos (deceased) for enabling and supporting the research on the Kulubnarti Nubians and thank Iosif Lazaridis for advice on analysis. K.A.S. was supported by a Doctoral Dissertation Research Improvement Grant from the National Science Foundation (BCS-1613577). D.R. was funded by NSF HOMINID grant BCS-1032255; NIH (NIGMS) grant GM100233; the Allen Discovery Center program, a Paul G. Allen Frontiers Group advised program of the Paul G. Allen Family Foundation; the John Templeton Foundation grant 61220; and the Howard Hughes Medical Institute.

## Author contributions

K.A.S., J.C.T. and D.V.G. conceived of the study and designed it with R.P. and D.R.; D.V.G. and R.P. facilitated sampling of skeletal material; C.S.H. performed radiocarbon analysis; K.A.S., N.R., N.A., A.M.L., K.C., M.F., M. Ma., E.H., R.B., N.B., M.F., M. Mi., J.O., F.Z. and K.S. performed ancient DNA laboratory and data processing work, and N.R. also supervised this work; K.A.S., D.M.F., M.L., S.M., M. Ma., I.O., H.R., E.H., N.P. analyzed genetic data. Supervision of this project was provided by N.P., J.C.T., D.V.G., R.P. and D.R.; K.A.S. wrote the manuscript with input from all authors.

## Competing interests

The authors declare no competing interests.
