## [Peer Review File · Nature Communications]

Title: Social stratification without genetic differentiation at the site of Kulubnarti in Christian Period NubiaREVIEWER COMMENTS

Reviewer #1 (Remarks to the Author):

Remarks to the authors

1. The article "Social stratification without genetic differentiation at the site of Kulubnarti in Christian Period Nubia" by Sirak et al. investigates the genetic history of an interesting region in northeast Africa, which has remained underrepresented in archaeogenetic research despite its critical importance to the understanding of human history and evolution. The skeletal material from Kulubnarti studied in this manuscript represents novel samples that date back to medieval Nubia. The number of aDNA samples analyzed another great success of the aDNA genetic technology in Africa, where DNA preservation, challenged by the harsh climate, deteriorates the recovery of sufficient amounts for sequencing. This work will pave the way for more studies from Africa, which will allow filling the gaps and answer big questions in our history.

2. The team of authors shows how can scientific research step forward through the efforts and proficiencies of researchers from different (anthropology, archeology, and genetics), but bridgeable disciplines.

3. The paper presented well-established molecular protocols and data analyses. It discusses the genetic results with great input from osteological and previous morphological findings and shows how genetics complements other fields, but most importantly how it helps revealing new insights into biological questions.

Here, I would like to address some comments.

Main manuscript:

1. Page 1, line 1: The title draws more attention to social stratification than to genetic intensive work, which is the main focus of the paper. I think it will look more attractive if the authors can highlight more about the genetic work achieved. The title weighs more on previous studies that pointed to the social stratification in the two cemeteries.

2. The introduction describes an interesting historical and archeological context with specific major questions about the biological relationships, that could not be resolved with morphological investigations only. Mostly, I enjoyed their description of the burial and grave styles and their cautiousness about different anthropological categories that could be jointly reported with genetic data.

3. The results section focuses mostly on the migration waves introduced into the region. The article has a pronounced and intensive analysis of the derived ancestry from the Middle East through Egypt. However, the results pointing to the link between modern-day Nubia and the samples from Kulubnarti

appeared in the last part of the results section. The article, apart from qpAdm and Dates analyses, didn't discuss much population continuity having both new aDNA data and already published modern data (cited in the article).

4. Page 14, line 483: The results section about the insights into modern-day Nubian people, was not very much surprised as one would expect. The modern-day data from Sudan is considerably appreciated and along with aDNA data can reveal more secrets about the ancestors of the modern-day people. Also, the continuity of gene flow into Nubia if maintained until the present day, we would expect to see higher levels of West-Eurasians mixtures in modern-day Nubians compared to genetic proportions in individuals who lived before Islamization of the region and the later influx of migrants.

5. An important addition to this article would be a simulation analysis through spatiotemporal dynamics of the populations using available data to better understand whether modern-day Nubia represents a continuum from ancient Nubia?

6. Page 3, line 100: The authors here describe the location of "Kulubnarti" as the measured distance from the Sudan-Egypt border, which in terms of political tension should be avoided. They could either say that "Kulubnarti" is located between the second and third cataracts, north of Khartoum, the capital of modern-day Sudan, or south of Wadi Halfa (123km), a city in the Northern state of Sudan. Having this said, the word Egypt outnumbered the word Sudan, both in the main text and Supplementary text. This might seem trivial, but with the increasing archeological appreciation of the field "Sudanology", this paper may better remain unbiased.

7. Page 5, line 187-188: "This is consistent with a scenario of fluidity between groups and the absence of a caste-like system of social division at Kulubnarti." Here I don't agree with the authors about the scenario of fluidity without taking into consideration other factors, such as the marriage system and external marriage relations and practices. They pointed to "exogamy" as a possible scenario when they discussed their results about female higher mobility from West Eurasian-related ancestry.

8. The authors could also discuss how the Egyptians were mediators of gene flow into Nubia utilizing linguistic affiliations, having radiocarbon dates that confirm the contemporaneity of the samples from Kulubnarti. How the Copts (Afro-Asiatic speakers) and the Nubians (Nilo-Saharan speakers) were connected? The importance of this region for trade connections cannot be studied without input from the culture and communications mediated through spoken languages. More discussion about how the Upper and Lower Nubia were connected in terms of linguistics will make a valuable addition to the anecdote of Medieval history.

9. The authors have well explained how they identified the outlier individuals (page 8, line 267-268). I was glad to see them reporting these outliers rather than removing them. However, I think these outlier individuals might represent a wave of new incomers from the south, owing to the excess of Nilotic ancestry. Or maybe original settlers at Kulubnarti were defeated by individuals of the larger size

community of Kulubnarti then.

10. I appreciated the wise decision in model fit analysis when the authors realized that the use of the Egypt_published was not a true proxy for the analysis. They could have also avoided using it in f3 statistics. I think the criticism received by the Egypt_published reference could also be avoided here by not reusing these samples to claim Middle eastern over sub-Saharan ancestry in the region.

11. The authors could visualize the results of qpAdm for the modern and aDNA data to ease reading the mixture patterns.

12. It is not clear or maybe not detailed enough in the methods or supplementary information how much Y and mtDNA coverage was achieved out of the 1240K? Were there any coverage differences among individuals that led only to the identification of diagnostic SNPs?

13. The new assigned mutations under the haplogroup U5b worth further investigation and probably with more aDNA samples researchers will be able to unlock the ancestral origins of the new sub-lineages.

14. It might be constructive to compare the female mediated gene flow results obtained from the model fit comparison between autosomes and the X-chromosome and the mtDNA haplogroup distribution among individuals at Kulubnarti.

15. Although this paper highlighted more about the back migration to Africa, it didn't discuss how the samples from Kulubnarti contributed to the gene pools of populations in the Arabian Peninsula and West Eurasia. This region of northeastern Africa is critical to our understanding of the Out of Africa (OoA) and multiregional hypotheses.

16. Last year, the American Society of Human Genetics (ASHG) published new professional guidance on ethical and legal considerations for ancient DNA genetics and genomics research (Wagner et al. 2020). I would like to know if the authors are aware of this professional guidance? And if so, how they prepared to engage with communities? Did they try to seek consultation with any of the entities that might be representative of the studied communities? What are the plans to communicate the results?

17. Page 4, line 138: remove and.

Supplementary Information:

1. The authors cited references for the dietary consumption at Kulubnarti based on isotopic data analyses, pointing to a diet based on seasonal plants. New research has identified milk proteins in

individuals from northeastern and eastern Africa through paleoproteomic. The work by Bleasdale and colleagues 2021 investigated sites from ancient Nubia spanning the Neolithic (~8000–5500 cal. BP) to Meroitic (~2300–1600 cal. BP) periods. The authors need not make clear-cut conclusive assumptions, knowing that other avenues of research such as paleoproteomic will help to uncover the unsolved questions of speculative hypotheses.

2. Supplementary Figure 2: The authors could add labels (a and b) to the figure and figure legend. This will make it easier to catch the difference between the reported data and expected patterns of ROH.

Reviewer #2 (Remarks to the Author):

Sirak et al report genome wide data for 66 individuals from the site of Kulubnarti from 650-1000 CE. The individuals show evidence of both Nilotic and WestEurasian ancestry but in varying proportions suggesting continuous gene-flow from the upper and lower part of the Nile respectively. This gene-flow also show evidence of being driven by female migrations.

Of particular interest is the fact that the individuals are buried in two different cemeteries. The remains in one cemetery (S cemetery) seems to be younger and have been subject to more stress compared to individuals buried at the R cemetery.

The authors find that the individuals do not cluster by ancestry, meaning that the individuals may have been socially divided but they are not genetically distinct.

Overall both the manuscript and supplementary material is very well written and the analysis are well documented - a pleasure to read!

However, there a couple of points that needs further investigation.

1) Table 1.

This table shows the number of within and across cemetery relationships and then gives a point estimate of how many inter- cemetery relationships would be expected if individuals were buried at random. The authors absolutely need to provide a range (in the form of 95% CI or similar). Otherwise you can't make statements like:

"While there are fewer inter-cemetery pairs of first- and second-degree relatives than expected if burials were completely random"

2) Female mediated gene-flow

This analysis is very interesting and the authors come up with a particular elegant way of estimating the magnitude of it, by running qpAdm on the X chromosome versus the autosomes.

I am a bit concerned as the X chromosome has much fewer SNPs compared to the autosomes. One way

to test the robustness of the analysis would be to run it on each autosome at a time (so first on chromosome 1, then 2, 3, and continue while there is a similar number of SNPs on the autosome and the X chromosome) and compare to the X chromosome. If there really is a difference between male and female migration patterns they should be true for each autosome in turn.

3) Continuous admixture

The authors write:

"Significant inter-individual variation in proportions of Nilotic- and West Eurasian-related ancestry suggests that admixture occurred relatively recently or was indeed ongoing".

If you use the word significant you need to show which statistical test was used and what the associated P-value is. This actually happens many times in the manuscript, please check each instance carefully.

When discussing if there is continuous or recent admixture the authors write:

"The inferred dates of admixture are significantly heterogeneous" - again which statistical test is used? In fact looking at figure 4 most (except R196) of the confidence intervals for admixture times overlap - suggesting that one admixture event could also explain a majority of the signal.

The authors need to show that they can distinguish these two scenarios.

I would suggest simulating admixture 22 generations ago and then simulating a few dozen individuals and down sampling then to get a similar coverage as you have in your data - this should tell you if you have the power to tell the scenarios apart.

Reviewer #3 (Remarks to the Author):

The manuscript presents the results of a well-designed genetic study aimed to investigate differences in ancestry in two contemporaneous Christian-period populations at Kulubnarti, northern Sudan. Previous archaeological and bioarchaeological studies of the two cemeteries established significant differences in morbidity and mortality between the two sub-groups interpreted as reflective of socioeconomic differences within the local population. Due to the limitations of morphological studies of skeletal remains, the question pertaining to biological relationship and ancestry of the Kulubnarti individuals (e.g. in relation to social stratification) remained open. By investigating palaeogenomic data obtained from the Kulubnarti individuals, with clearly formulated research questions, Authors were able to establish the following:

(a) genetically related individuals were buried in both cemeteries, thus genetic differentiation was not a factor in social stratification

(b) the Kulubnarti individuals were admixed on average with ~43% Nilotic-related ancestry and the remaining

ancestry reflected a West Eurasian-related gene pool likely introduced into Nubia through Egypt from Levant

(c) the West Eurasian-related ancestry derived from female ancestors.

These findings are highly significant and provide a new insight into the Christian population of Kulubnarti, its genetic and social structure, as well as mobility and genetic dynamics in the broader region through time. By identifying the limitations of commonly applied research methods and resulting interpretations, this study clearly demonstrates the value and importance of incorporating genetic studies in investigating past populations.

The research is well designed and clearly presented. The research questions are clearly defined and background information is given in detail. The methodology is appropriate for the study and presented in detail. The findings (palaeogenomic data) are of great significance for the region. I recommend the manuscript to be published.

Reviewer #4 (Remarks to the Author):

I review this manuscript as an archaeological scientist and radiocarbon dating expert, focusing on areas within my expertise.

Overall, I find this paper to provide evidence for its conclusions and of interest to workers within the field and related disciplines. The genetic evidence is well integrated with archaeological data, with a comprehensive discussion of the findings.

Recommendations:

For figure 1, I suggest you edit panel C to to enlarge your OxCal output. Particularly for print, it is best if the font style and size within a figure match.

Although it would have been best practice to run the cemeteries separately in OxCal rather than in a single, overlapping Phase (since distinct archaeological sites), the Bayesian modelling is generally sound. Moreover, when I ran each site separately with an Outlier Analysis (General type), my results and this manuscript's are statistically indistinguishable, with the distributions overlapping zero at 95% probability.

Since an EA-IRMS was used on the bone collagen samples that were radiocarbon dated, C%, N% and d15N values should be reported in Supplementary Table 1. Combined with the collagen yield, CN ratio and d13C, these are quality assurance parameters that allow us to assess the reliability of radiocarbon dates reported. On this note, there are seven bone samples with high CN ratios (>3.5), which likely indicate contamination (van Klinken 1999). Sample S208, in particular, has a value of 4.8. These radiocarbon measurements cannot be considered to be accurate/reliable (there are laboratories that would not report them). If collagen is left, they should be ultrafiltered (routine in collage dating) to further decontaminate and duly reassessed. Otherwise, at the very least, this needs to be discussed within the paper and the dates flagged.

References

Van Klinken GJ. 1999. Bone collagen quality indicators for palaeodietary and radiocarbon measurements. *Journal of Archaeological Science*. 26(6):687-695.

REVIEWER COMMENTS

Reviewer #1 (Remarks to the Author):

Remarks to the authors

1. The article "Social stratification without genetic differentiation at the site of Kulubnarti in Christian Period Nubia" by Sirak et al. investigates the genetic history of an interesting region in northeast Africa, which has remained underrepresented in archaeogenetic research despite its critical importance to the understanding of human history and evolution. The skeletal material from Kulubnarti studied in this manuscript represents novel samples that date back to medieval Nubia. The number of aDNA samples analyzed another great success of the aDNA genetic technology in Africa, where DNA preservation, challenged by the harsh climate, deteriorates the recovery of sufficient amounts for sequencing. This work will pave the way for more studies from Africa, which will allow filling the gaps and answer big questions in our history.
2. The team of authors shows how can scientific research step forward through the efforts and proficiencies of researchers from different (anthropology, archeology, and genetics), but bridgeable disciplines.
3. The paper presented well-established molecular protocols and data analyses. It discusses the genetic results with great input from osteological and previous morphological findings and shows how genetics complements other fields, but most importantly how it helps revealing new insights into biological questions.

Here, I would like to address some comments.

Main manuscript:

1. Page 1, line 1: The title draws more attention to social stratification than to genetic intensive work, which is the main focus of the paper. I think it will look more attractive if the authors can highlight more about the genetic work achieved. The title weighs more on previous studies that pointed to the social stratification in the two cemeteries.

We reflected upon this comment, and we feel that the title "Social stratification without genetic differentiation at the site of Kulubnarti in Christian Period Nubia" reflects a key finding of this work, which was primarily inspired by the opportunity to test the hypothesis of socioeconomic differences and biological similarity at Kulubnarti. Arguably the most critical genetic finding is highlighted, namely that there are no significant genetic differences between the two cemeteries. Furthermore, we would like to retain this title as it highlights the multidisciplinary nature of this work, which uses a new line of evidence to explore a question that arose from decades of archaeological and anthropological analyses.

2. The introduction describes an interesting historical and archeological context with specific major questions about the biological relationships, that could not be resolved with morphological investigations only. Mostly, I enjoyed their description of the burial and grave styles and their cautiousness about different anthropological categories that could be jointly reported with genetic data.
3. The results section focuses mostly on the migration waves introduced into the region. The article has a pronounced and intensive analysis of the derived ancestry from the Middle East through Egypt. However, the results pointing to the link between modern-day Nubia and the samples from Kulubnarti appeared in the last part of the results section. The article, apart from qpAdm and Dates analyses, didn't discuss much

population continuity having both new aDNA data and already published modern data (cited in the article).

In our revised manuscript, we provide even more evidence than in our original submission that we find no evidence of population continuity between the Christian Period people from Kulubnarti and the present-day Nubian groups published in Hollfelder et al. 2017 (Mahas, Halfawieen, and Danagla).

We begin by reviewing the analyses already presented in our original submission. First, we used qpWave to show that none of these present-day Nubian groups form a clade with the Kulubnarti Nubians, implying that they are not descended from the same ancestral populations without additional admixture. Second, we tested if the same proxy source populations for the ancestry of the Kulubnarti Nubians could be applied to present-day Nubian groups using qpAdm, but found that they could not. Third, we estimated the time of admixture that contributed to modern Nubian groups, and found that these dates support the conclusion made in Hollfelder et al. 2017 that the Arab migrations which introduced Islam into the Nile Valley region also left a genetic impact that is still detectable in present-day Nubian groups.

We have now added an additional analysis for this resubmission, using qpAdm to test whether the Kulubnarti Nubians fit as a source population for ancestry in any one of the present-day Nubian groups. Using our same O9 + Anatolia_EBA reference population set that we used to estimate ancestry in the Kulubnarti Nubians, we attempt to model the three present-day Nubian groups as mixtures of the Kulubnarti Nubians plus Dinka. We find no valid models using Kulubnarti as a source population for any of the three present-day Nubian groups. We present these new results in Supplementary Data 7, and have added the following sentence into the main text: "...we were also not able to fit Kulubnarti as a source in a two-way admixture model for any present-day Nubian group ($p < 2.1E-06$)."

A lack of clearly detectable population continuity between the Christian Period and the present day is consistent with the historical and material cultural records, which show substantial changes introduced after the Christian Period, and which also suggests that the Kulubnarti Nubians may have inherited a contribution of West Eurasian-related ancestry (perhaps related to Christians coming from the north), that did not contribute much to later groups.

We emphasize that the Kulubnarti Nubians are only a single group and we have data from only three present-day Nubian populations for comparison. It is of course possible and even likely that future work with increased amounts of data from additional ancient and present-day Nubian populations will identify ancestry from other ancient groups that is still detectable in present-day Nubians.

4. Page 14, line 483: The results section about the insights into modern-day Nubian people, was not very much surprised as one would expect. The modern-day data from Sudan is considerably appreciated and along with aDNA data can reveal more secrets about the ancestors of the modern-day people. Also, the continuity of gene flow into Nubia if maintained until the present day, we would expect to see higher levels of West-Eurasians mixtures in modern-day Nubians compared to genetic proportions in individuals who lived before Islamization of the region and the later influx of migrants.

The dates of admixture for present-day Nubians that we estimate with the DATES method suggests that it is likely that the migrations of people of West Eurasian-related ancestry who spread Islam throughout Nubia had a genetic impact. It is also possible that there have been additional waves of migration of people carrying other types of ancestry, such as Nilotic-related ancestry. Indeed, Hollfelder et al. 2017 (the paper that initially published these present-day data) note that "...the Nubians can be seen as a group with substantial genetic material relating to Nilotes that later have received much gene flow from Eurasians (likely Middle Eastern) and from East Africans." Future research will contribute to a greater understanding of the genetic changes in Nubian populations over time.

5. An important addition to this article would be a simulation analysis through spatiotemporal dynamics

of the populations using available data to better understand whether modern-day Nubia represents a continuum from ancient Nubia?

We believe that the analyses that we use in this paper (including qpWave, qpAdm, and estimation of admixture dates) clearly indicate that the people of the three present-day Nubian groups for whom we have genotype data did not descend directly from the Christian Period group at Kulubnarti without additional waves of admixture. With our single ancient data point from Kulubnarti and three modern datapoints (the three population samples from Hollfelder et al. 2017), we do not think that we have enough spatial, temporal, and geographic variation to make a simulation series add meaningfully to the analyses we already carry out.

6. Page 3, line 100: The authors here describe the location of “Kulubnarti” as the measured distance from the Sudan-Egypt border, which in terms of political tension should be avoided. They could either say that “Kulubnarti” is located between the second and third cataracts, north of Khartoum, the capital of modern-day Sudan, or south of Wadi Halfa (123km), a city in the Northern state of Sudan. Having this said, the word Egypt outnumbered the word Sudan, both in the main text and Supplementary text. This might seem trivial, but with the increasing archeological appreciation of the field “Sudanology”, this paper may better remain unbiased.

We have changed the description of Kulubnarti’s location in the main text and supplementary text. The main text now reads: “Here we present genome-wide data from 66 individuals who lived at Kulubnarti, located between the second and third cataracts of the Nile approximately 120 kilometers south of the Sudanese city of Wadi Halfa.” The supplementary text now reads: “The site of Kulubnarti (“Island of Kulb” in the Mahasi dialect of Nubian) is located on the bank of the Nile River in Sudanese Nubia, approximately 120 kilometers south of the present-day Sudanese city of Wadi Halfa.”

7. Page 5, line 187-188: “This is consistent with a scenario of fluidity between groups and the absence of a caste-like system of social division at Kulubnarti.” Here I don’t agree with the authors about the scenario of fluidity without taking into consideration other factors, such as the marriage system and external marriage relations and practices. They pointed to “exogamy” as a possible scenario when they discussed their results about female higher mobility from West Eurasian-related ancestry.

Our genetic data – specifically the overall signal of genetic similarity between individuals buried in the R and S cemeteries and especially the identification of related individuals buried in either cemetery – suggests that there were not significant genetic differences between individuals buried in the Kulubnarti cemeteries that would be consistent with social system that forbade mixing between groups. Whether mixing between groups was something that took place through a system of marriage or extra-marital unions cannot be deciphered through genetic data alone, and anthropological data does not provide a clear indication of the mechanism by which gene flow between these two plausibly stratified groups occurred. As such, we can only speak to the interpretations that we can draw from our genetic data, which shows no absence of gene flow. In response to the reviewer’s comment and for additional clarity, we have reworded this to read: “This is consistent with a scenario where any system of social division at Kulubnarti did not prevent gene flow between plausibly stratified groups.”

8. The authors could also discuss how the Egyptians were mediators of gene flow into Nubia utilizing linguistic affiliations, having radiocarbon dates that confirm the contemporaneity of the samples from Kulubnarti. How the Copts (Afro-Asiatic speakers) and the Nubians (Nilo-Saharan speakers) were connected? The importance of this region for trade connections cannot be studied without input from the culture and communications mediated through spoken languages. More discussion about how the Upper and Lower Nubia were connected in terms of linguistics will make a valuable addition to the anecdote of Medieval history.

We have included a sentence recognizing the presence of Christian Period inscriptions at Kulubnarti which are in Greek and Coptic as well as Old Nubian, specifically noting that the substantial proportion of West Eurasian-related ancestry found in the Kulubnarti Nubians is "...consistent with evidence of West Eurasian and Egyptian influence at Kulubnarti, including Christian churches as well as inscriptions in Greek and Coptic as well as Old Nubian." We also have added the statement: "It has even been suggested that the Kulubnarti Nubians could have migrated into the Batn el Hajar from the north", as mentioned in Adams et al. 1999. We look forward to making the genetic data from this work fully public so that it can be used in the future to more clearly illuminate the cultural and biological connections between the people of the Nile Valley.

9. The authors have well explained how they identified the outlier individuals (page 8, line 267-268). I was glad to see them reporting these outliers rather than removing them. However, I think these outlier individuals might represent a wave of new incomers from the south, owing to the excess of Nilotic ancestry. Or maybe original settlers at Kulubnarti were defeated by individuals of the larger size community of Kulubnarti then.

In this work, "outliers" denotes an individual with ancestry proportions that are highly significantly distant from the group mean rather than an individual who is part of a distinct genetic cluster (we note this in the main text). While it is possible that the outlier individuals represent new migrants to Kulubnarti, this is unclear given the observed pattern of overly-dispersed heterogeneity in terms of proportion of Nilotic- and West Eurasian-related ancestry among the Kulubnarti Nubians—present even among the non-outliers—consistent with a scenario of relatively recent or ongoing admixture. Indeed, the signal of continuing admixture is also evident through the significantly heterogeneous single-sample admixture dates. Future work, including heavy isotope analysis (such as $^{87}\text{Sr}/^{86}\text{Sr}$), may be helpful in determining whether these individuals originate from elsewhere.

10. I appreciated the wise decision in model fit analysis when the authors realized that the use of the Egypt_published was not a true proxy for the analysis. They could have also avoided using it in f_3 statistics. I think the criticism received by the Egypt_published reference could also be avoided here by not reusing these samples to claim Middle eastern over sub-Saharan ancestry in the region.

Our ancestry modeling using qpAdm suggests that a population related to ancient Egyptians are the best proxy for the proximal source of the West Eurasian-related ancestry found at Kulubnarti; however, as noted by this reviewer, it is clear that the source of this ancestry ultimately lies outside of Africa (and our analyses indicate that it most likely originates in the Bronze and Iron Age Levant, a finding in line with previous research). The Egypt_published population is used in f_3 statistics as a proxy for West Eurasian-related ancestry, as modelling suggests that they have about 95% Levant Bronze/Iron Age-related ancestry, and as analyses in our paper do make it clear that they have distinctive connections to Kulubnarti above and beyond those in ancient West Eurasians.

11. The authors could visualize the results of qpAdm for the modern and aDNA data to ease reading the mixture patterns.

We have added a figure showing the qpAdm results for each individual at Kulubnarti as Supplementary Fig. 5.

12. It is not clear or maybe not detailed enough in the methods or supplementary information how much Y and mtDNA coverage was achieved out of the 1240K? Were there any coverage differences among individuals that led only to the identification of diagnostic SNPs?

We added a column (column C) in Supplementary Data 13 that provides the number of SNPs covered on the Y chromosome for the 30 males reported in this work.

mtDNA coverages for each individual are provided in Supplementary Data 2 (column M); in addition, the rate of matching to the mtDNA consensus sequence and the 95% CI are provided in columns O and P.

13. The new assigned mutations under the haplogroup U5b worth further investigation and probably with more aDNA samples researchers will be able to unlock the ancestral origins of the new sub-lineages.

We agree that further investigations of the origins of this haplogroup is of interest and will be aided by more ancient DNA data that will hopefully be generated in the future.

14. It might be constructive to compare the female mediated gene flow results obtained from the model fit comparison between autosomes and the X-chromosome and the mtDNA haplogroup distribution among individuals at Kulubnarti.

We address this in our manuscript. We find that 35 individuals at Kulubnarti who belong to mtDNA haplogroups with origins in West Eurasia is in the range of expected for anywhere between 43–68% of maternal ancestry at Kulubnarti coming from West Eurasian ancestors (based on evaluating whether each proportion in this range included 35 West Eurasian mitochondrial haplogroups within its 95% central confidence interval). This overlaps the 59–77% estimate of West Eurasian-related ancestry deriving from females made by comparing ancestry proportions on the autosomes and X chromosome.

15. Although this paper highlighted more about the back migration to Africa, it didn't discuss how the samples from Kulubnarti contributed to the gene pools of populations in the Arabian Peninsula and West Eurasia. This region of northeastern Africa is critical to our understanding of the Out of Africa (OoA) and multiregional hypotheses.

As we do not analyze ancient individuals from the Arabian Peninsula or West Eurasia in this work, which is focused on exploring the genetic history of the population from Kulubnarti, we believe this exploration is outside of our scope of study.

In addition, the fact that one of the important findings in our paper is that the Kulubnarti Nubians have received distinctive West Eurasian-related admixture within hundreds of years of the time they lived means that they cannot be viewed as a plausible proxy source for out-of-Africa migrations. The question of the sources of these migrations is of course extremely important, and could potentially be explored through older successful ancient DNA analysis from the region.

16. Last year, the American Society of Human Genetics (ASHG) published new professional guidance on ethical and legal considerations for ancient DNA genetics and genomics research (Wagner et al. 2020). I would like to know if the authors are aware of this professional guidance? And if so, how they prepared to engage with communities? Did they try to seek consultation with any of the entities that might be representative of the studied communities? What are the plans to communicate the results?

We are aware of the professional guidance issued by the ASHG for ethical and legal considerations of ancient DNA genetics and genomics research in a US context (as articulated in Wagner et al. 2020).

We have now added an Ethics Statement in the main text to describe how ethical considerations were taken into account in a dedicated way in our work:

“We acknowledge the ancient individuals whose remains we analyzed in this work and who must be treated with respect. Excavation of human remains from the two Kulubnarti cemeteries occurred in 1979 under a license granted by the Sudan Antiquities Service (now the National Corporation for Antiquities and Museums) to Dr. William Y. Adams; the excavation of the cemeteries was funded by the National Science Foundation (Grant No. 77-270210-535), and led by Dr. Dennis Van Gerven, co-senior author on this work; it was undertaken as a part of the UNESCO International Campaign to Save the Monuments of Nubia. Prior to the excavation of the Kulubnarti cemeteries, the head of the Sudanese Antiquities Service

approved the research plan, including the invasive investigations (such as biochemical analyses) anticipated at the time. All graves to be excavated were marked by the archaeological team and their excavation was approved by the Nubian reiss (foreman). The excavation was inspected by a representative of the Sudan Antiquities Service monthly, and all excavated remains were reinspected prior to their export, which occurred in accordance with the regulations at that time. At the time of excavation, the people living around Kulubnarti did not identify remains in Christian-style graves as their ancestors and communicated this to Dr. Van Gerven and his colleagues. Based on the wishes of the local community, Dr. Van Gerven's team avoided any interference with Muslim graves in the Kulubnarti cemeteries, which were distinguishable by their north-south orientation."

The first author has made arrangements to carry out a dedicated return of results to both the academic community and to the public in Sudan. If it is safe and responsible to travel, the first author will return to Sudan in November/December 2021 and discuss ancient DNA research in general and these results in particular in three forums: first, at the National Corporation for Antiquities and Museums (NCAM, who are in charge of protecting and maintaining the archaeological heritage of Sudan); second, at the University of Khartoum, where there are students studying archaeology and bioanthropology; and finally, to the general public, likely at one of the foreign institutes located centrally in Khartoum.

17. Page 4, line 138: remove and.

We have made this correction.

Supplementary Information:

1. The authors cited references for the dietary consumption at Kulubnarti based on isotopic data analyses, pointing to a diet based on seasonal plants. New research has identified milk proteins in individuals from northeastern and eastern Africa through paleoproteomic. The work by Bleasdale and colleagues 2021 investigated sites from ancient Nubia spanning the Neolithic (~8000–5500 cal. BP) to Meroitic (~2300–1600 cal. BP) periods. The authors need not make clear-cut conclusive assumptions, knowing that other avenues of research such as paleoproteomic will help to uncover the unsolved questions of speculative hypotheses.

We cite the published body of literature that uses isotopic information to explore dietary patterns in the Kulubnarti Nubians and look forward to additional discoveries made through paleoproteomics work.

2. Supplementary Figure 2: The authors could add labels (a and b) to the figure and figure legend. This will make it easier to catch the difference between the reported data and expected patterns of ROH.

We have added labels for panels a and b as suggested.

Reviewer #2 (Remarks to the Author):

Sirak et al report genome wide data for 66 individuals from the site of Kulubnarti from 650-1000 CE. The individuals show evidence of both Nilotic and WestEurasian ancestry but in varying proportions suggesting continuous gene-flow from the upper and lower part of the Nile respectively. This gene-flow also show evidence of being driven by female migrations.

Of particular interest is the fact that the individuals are buried in two different cemeteries. The remains in one cemetery (S cemetery) seems to be younger and have been subject to more stress compared to

individuals buried at the R cemetery.

The authors find that the individuals do not cluster by ancestry, meaning that the individuals may have been socially divided but they are not genetically distinct.

Overall both the manuscript and supplementary material is very well written and the analysis are well documented - a pleasure to read!

However, there are a couple of points that need further investigation.

1) Table 1.

This table shows the number of within and across cemetery relationships and then gives a point estimate of how many inter-cemetery relationships would be expected if individuals were buried at random. The authors absolutely need to provide a range (in the form of 95% CI or similar). Otherwise you can't make statements like:

"While there are fewer inter-cemetery pairs of first- and second-degree relatives than expected if burials were completely random"

We have revised this analysis in response to the Referee's suggestion. We now included an updated Table 1 in which we compute the expected probability that two randomly chosen individuals of known-degree relatedness at Kulubnarti are cross-cemetery relatives, and compute the expected number of cross-cemetery relative pairs. We then determine the rate of observed pairs as a percentage of the expectation, and compute a p-value from a binomial distribution of the probability that the reduction in cross-cemetery relative pairs is significant. We then report a p-value for this in the text. To describe this analysis, we write: "Comparing whether 32 individuals with relatives of a known-degree in our dataset were part of within- or across-cemetery relative pairs, we find a significant reduction in the expected rate of cross-cemetery relatives if cemetery of burial exhibited no correlation to family structure when we pool all first- and second-degree and all first-, second-, and third-degree relative pairs (both $p=0.021$), though this effect is not significant when relative pairs of each degree are analyzed separately (all $p>0.105$). The rate of cross-cemetery relative pairs relative to expectation is 0% for first-degree relative pairs (0 observed compared to 1.7 expected), 46% for second-degree relative pairs (2 observed compared to 4.37 expected), and 95% for third-degree relative pairs (4 observed compared to 4.21 expected). These results show that there is some enrichment of relative pairs buried in the same cemetery versus in different cemeteries which is most evident for first- and second-degree relative pairs (Table 1), as expected if people tended to be buried with very close family members. However, the signal attenuates to a level that is indistinguishable from random at the third-degree relative level. This is consistent with a scenario where any system of social division at Kulubnarti did not prevent gene flow between plausibly stratified groups."

2) Female mediated gene-flow

This analysis is very interesting and the authors come up with a particular elegant way of estimating the magnitude of it, by running qpAdm on the X chromosome versus the autosomes. I am a bit concerned as the X chromosome has much fewer SNPs compared to the autosomes. One way to test the robustness of the analysis would be to run it on each autosome at a time (so first on chromosome 1, then 2, 3, and continue while there is a similar number of SNPs on the autosome and the X chromosome) and compare to the X chromosome. If there really is a difference between male and female migration patterns they should be true for each autosome in turn.

Running the analysis in large contiguous chunks of the genome at a time (such as each chromosomes in turn, or in practice 5 centimorgan chunks of the autosomes and X chromosomes) is effectively what we do in order to infer standard errors: this is the source of our Block Jackknife standard errors. As these standard errors are used for the Z-score calculation, the result can be confidently interpreted as indicating a statistically significant signal.

In order to explicitly follow the reviewer's suggestion, we ran qpAdm on one autosomal chromosome one at a time and compared the results to the X chromosome results as suggested. We find that in every case, a positive Z-score indicates greater West Eurasian-related ancestry on the X chromosome than any chromosome. While for some chromosomes the Z-score falls under the significance threshold of $|Z| > 3$ due to a relatively high standard error for both the autosome and X chromosome, the results are qualitatively consistent with our finding presented in the manuscript, which leverages increased statistical power from combining the autosomes which reduces the standard errors. We do not add this table to the main text as the Block Jackknife is based on exactly this kind of analysis and so such an analysis would be redundant.

Chr	Autosomes						X chromosome						Z-score
	Estimated proportion		Standard Error		P-value	SNPs	Estimated proportion		Standard Error		P-value	SNPs	
	Dinka	Levant BAIA	Dinka	Levant BAIA			Dinka	Levant BAIA	Dinka	Levant BAIA			
1	44.30%	55.70%	1.20%	1.20%	0.348	60459	35.6%	64.4%	1.8%	1.8%	0.988	19577	4.02
2	41.70%	58.30%	1.10%	1.10%	0.987	62612	35.6%	64.4%	1.8%	1.8%	0.988	19577	2.89
3	42.50%	57.50%	1.50%	1.50%	0.102	51653	35.6%	64.4%	1.8%	1.8%	0.988	19577	2.94
4	39.30%	60.70%	1.40%	1.40%	0.306	43656	35.6%	64.4%	1.8%	1.8%	0.988	19577	1.62
5	43.90%	56.10%	1.40%	1.40%	0.306	46805	35.6%	64.4%	1.8%	1.8%	0.988	19577	3.64
6	41.70%	58.30%	1.10%	1.10%	0.053	49337	35.6%	64.4%	1.8%	1.8%	0.988	19577	2.89
7	42.70%	57.30%	1.30%	1.30%	0.589	39070	35.6%	64.4%	1.8%	1.8%	0.988	19577	3.20
8	42.70%	57.30%	1.20%	1.20%	0.838	40428	35.6%	64.4%	1.8%	1.8%	0.988	19577	3.28
9	41.10%	58.90%	1.50%	1.50%	0.213	33388	35.6%	64.4%	1.8%	1.8%	0.988	19577	2.35
10	40.30%	59.70%	1.90%	1.90%	0.211	39435	35.6%	64.4%	1.8%	1.8%	0.988	19577	1.80
11	44.80%	55.20%	1.10%	1.10%	0.424	36615	35.6%	64.4%	1.8%	1.8%	0.988	19577	4.36
12	40.50%	59.50%	1.00%	1.00%	0.306	35589	35.6%	64.4%	1.8%	1.8%	0.988	19577	2.38
13	42.60%	57.40%	1.60%	1.60%	0.018	24690	35.6%	64.4%	1.8%	1.8%	0.988	19577	2.91
14	40.80%	59.20%	1.20%	1.20%	0.18	24036	35.6%	64.4%	1.8%	1.8%	0.988	19577	2.40
15	44.10%	55.90%	1.10%	1.10%	0.03	23326	35.6%	64.4%	1.8%	1.8%	0.988	19577	4.03
16	41.10%	58.90%	1.70%	1.70%	0.075	23871	35.6%	64.4%	1.8%	1.8%	0.988	19577	2.22
17	43.00%	57.00%	1.50%	1.50%	0.416	20578	35.6%	64.4%	1.8%	1.8%	0.988	19577	3.16
18	41.10%	58.90%	1.30%	1.30%	0.082	22342	35.6%	64.4%	1.8%	1.8%	0.988	19577	2.48
19	39.80%	60.20%	1.20%	1.20%	0.642	12817	35.6%	64.4%	1.8%	1.8%	0.988	19577	1.94
20	42.60%	57.40%	1.20%	1.20%	0.067	20183	35.6%	64.4%	1.8%	1.8%	0.988	19577	3.24
21	35.80%	64.20%	1.50%	1.50%	2.65E-10	10430	35.6%	64.4%	1.8%	1.8%	0.988	19577	0.09
22	41.90%	50.90%	2.00%	2.00%	0.002	11082	35.6%	64.4%	1.8%	1.8%	0.988	19577	2.34

3) Continuous admixture

The authors write:

"Significant inter-individual variation in proportions of Nilotic- and West Eurasian-related ancestry suggests that admixture occurred relatively recently or was indeed ongoing".

If you use the word significant you need to show which statistical test was used and what the associated P-value is. This actually happens many times in the manuscript, please check each instance carefully.

We have gone through the manuscript and replaced the word significant in cases where we make a qualitative statement.

When discussing if there is continuous or recent admixture the authors write:

"The inferred dates of admixture are significantly heterogeneous" - again which statistical test is used? In fact looking at figure 4 most (except R196) of the confidence intervals for admixture times overlap - suggesting that one admixture event could also explain a majority of the signal.

The authors need to show that they can distinguish these two scenarios.

I would suggest simulating admixture 22 generations ago and then simulating a few dozen individuals and down sampling then to get a similar coverage as you have in your data - this should tell you if you have the power to tell the scenarios apart.

We used the phrase "significantly heterogeneous" in our initial submission to describe the admixture date estimates presented in Figure 4, which displays both the point estimates and the 95% confidence intervals determined using the standard deviation provided by the DATES software; a Z-score >2.8 (corresponding to a 99.5% CI) for the individual estimate and a direct ¹⁴C date was required for inclusion in this figure. Using this conservative approach, we find that there are individuals from Kulubnarti with 95% CI intervals that do not overlap, although we agree that this does not correct for the number of individuals (hypotheses) we tested.

We appreciate the reviewer's concern about the ability to distinguish between admixture scenarios. The software that we use in this work (DATES) measures the decay of ancestry covariance to infer the time since admixture occurred. A limitation is that the dates obtained are based on a model of a single pulse of admixture and thus reflect an intermediate value if the true population history includes multiple waves of admixture or continuous admixture (which is likely at Kulubnarti as evidenced by individual-level variation in ancestry proportions). It is not possible with this method to determine specifically how many waves of admixture contributed to the gene pool at Kulubnarti. For this reason, we exercise great caution in discussing our results, including noting in two places in the main text that values reflect an average if there were multiple waves or continuous admixture (which is likely), and explaining that our non-overlapping 95% CI estimates confirm that waves of admixture over the time period on the order of a millennium could have contributed to the Kulubnarti gene flow. As methods for estimating the number and dates of admixture pulses in the past improve, we expect to better refine this finding.

To address the Referee's suggestion, now added any analysis of statistical significance between all DATES estimates, with the analysis described and data provided in Supplementary Table 5. After Bonferroni correction for multiple hypotheses (which raises the significance threshold to $|Z|=3.65$), we find a statistically significant difference between estimated dates of admixture for S133 and S73 ($Z=3.9$) and S133 and R196 ($Z=4.1$). We describe this new analysis in the revised paragraph, which now reads: "Though most pairs of individuals have overlapping inferred admixture dates (95% CI), this is not true of every pair, providing evidence that the admixture did not all occur at a single time. Considering the twenty individuals who also have ¹⁴C dates as well as DATES estimates allows us additional insight into the timing of admixture. Here, using calibrated ¹⁴C dates, we observe point estimates ranging from ~200 BCE (95% CI, ~490 BCE–100 CE) to ~660 CE (95% CI, 470–850 CE). We computed Z-scores for the difference between all possible pairs (Supplementary Table 5) and found significant variation in inferred

admixture dates after Bonferroni correction for significance based on the number of hypotheses tested ($|Z| > 3.65$), confirming that there is significant variation in average admixture date estimates and suggesting that waves of admixture extending on the order of a millennium contributed to the formation of the Kulubnarti gene pool.”

Reviewer #3 (Remarks to the Author):

The manuscript presents the results of a well-designed genetic study aimed to investigate differences in ancestry in two contemporaneous Christian-period populations at Kulubnarti, northern Sudan. Previous archaeological and bioarchaeological studies of the two cemeteries established significant differences in morbidity and mortality between the two sub-groups interpreted as reflective of socioeconomic differences within the local population. Due to the limitations of morphological studies of skeletal remains, the question pertaining to biological relationship and ancestry of the Kulubnarti individuals (e.g. in relation to social stratification) remained open. By investigating palaeogenomic data obtained from the Kulubnarti individuals, with clearly formulated research questions, Authors were able to establish the following:

(a) genetically related individuals were buried in both cemeteries, thus genetic differentiation was not a factor in social stratification

(b) the Kulubnarti individuals were admixed on average with ~43% Nilotic-related ancestry and the remaining ancestry reflected a West Eurasian-related gene pool likely introduced into Nubia through Egypt from Levant

(c) the West Eurasian-related ancestry derived from female ancestors.

These findings are highly significant and provide a new insight into the Christian population of Kulubnarti, its genetic and social structure, as well as mobility and genetic dynamics in the broader region through time. By identifying the limitations of commonly applied research methods and resulting interpretations, this study clearly demonstrates the value and importance of incorporating genetic studies in investigating past populations.

The research is well designed and clearly presented. The research questions are clearly defined and background information is given in detail. The methodology is appropriate for the study and presented in detail. The findings (palaeogenomic data) are of great significance for the region. I recommend the manuscript to be published.

We thank the reviewer for these positive comments.

Reviewer #4 (Remarks to the Author):

I review this manuscript as an archaeological scientist and radiocarbon dating expert, focusing on areas within my expertise.

Overall, I find this paper to provide evidence for its conclusions and of interest to workers within the field and related disciplines. The genetic evidence is well integrated with archaeological data, with a comprehensive discussion of the findings.

Recommendations:

For figure 1, I suggest you edit panel C to to enlarge your OxCal output. Particularly for print, it is best if the font style and size within a figure match.

We have edited Figure 1 panel c.

Although it would have been best practice to run the cemeteries separately in OxCal rather than in a single, overlapping Phase (since distinct archaeological sites), the Bayesian modelling is generally sound. Moreover, when I ran each site separately with an Outlier Analysis (General type), my results and this manuscript's are statistically indistinguishable, with the distributions overlapping zero at 95% probability.

In response to the Referee's suggestion, we have re-run the model as two separate independent phases. We also now include the general outlier model, which better accounts for possible variability related to potential contaminants although the results are statistically indistinguishable.

Since an EA-IRMS was used on the bone collagen samples that were radiocarbon dated, C%, N% and d15N values should be reported in Supplementary Table 1. Combined with the collagen yield, CN ratio and d13C, these are quality assurance parameters that allow us to assess the reliability of radiocarbon dates reported. On this note, there are seven bone samples with high CN ratios (>3.5), which likely indicate contamination (van Klinken 1999). Sample S208, in particular, has a value of 4.8. These radiocarbon measurements cannot be considered to be accurate/reliable (there are laboratories that would not report them). If collagen is left, they should be ultrafiltered (routine in collage dating) to further decontaminate and duly reassessed. Otherwise, at the very least, this needs to be discussed within the paper and the dates flagged.

We have added %C, %N, and d15N values to Supplementary Table 1. Samples with C:N ratios outside the accepted range have been flagged in the Supplementary Table. However, they are included in the models, and a general outlier model applied to down-weight possible outliers.

We appreciate that ultrafiltration of contaminated collagen can provide more reliable age estimates in some cases, particularly for samples over 2-3 half-lives in age (Higham et al. 2013), or those with lower (<3%) collagen yields (Minami et al. 2013). However, the literature on the use of ultrafiltration highlights the risks, as well as the potential benefits, of this added step. Brock et al. (2007), Hüls et al. (2007, 2009), and Minami et al (2013) have shown through controlled experiments that the use of ultrafilters introduces contaminating carbon from the filter and its humectant, despite following a rigorous cleaning protocol on the ultrafilters themselves. Furthermore, ultrafilters cannot remove (and can possibly concentrate) high-molecular-weight collagen cross-linked with humic acid contaminants.

Considering the "young" age of the samples (significantly less than 1 half-life), the large collagen yields (and only one sample could be considered to have a low collagen yield at 2.8%), and the potential for contamination related to the use of ultrafilters themselves, we opted for the routine Longin-type extraction without ultrafiltration. We opted to flag the dates with aberrant C:N ratios, and adopted a general outlier model in the Bayesian chronological model.

REVIEWER COMMENTS

Reviewer #1 (Remarks to the Author):

The revised article "Social stratification without genetic differentiation at the site of Kulubnarti in Christian Period Nubia" by Sirak et al. has covered precisely all the comments raised by the reviewers. I would like to thank the authors for their clear responses and for addressing the answers in the revised version.

I am glad to see that an Ethics statement was added to the results section. This statement weighs a lot on this well-written article and its novel data.

I would like to thank the authors for their responses to some comments that were probably raised because of unclarity in the original submission. In the revised version they have very well-considered and addressed them to help their readers avoid confusion. This will, without a doubt, attract the readers to the novelty of the data and the analysis conducted to provide pieces of evidence.

I am optimistic that the step taken by the authors to share their results with the communities and/or community representatives will allow aDNA studies to receive appreciation from non-scientific communities and the public.

Reviewer #2 (Remarks to the Author):

The authors have now updated their analysis and I have no further comments for the scientific analysis.

However based on the response to reviewers 1's question regarding the guidelines from the American Society of Human Genetics (ASHG) I have other concerns! The main points listed in Wagner et al 2020 are:

- (1) formally consult with communities
- (2) address cultural and ethical considerations
- (3) engage communities and support capacity building
- (4) develop plans to report results and manage data
- (5) develop plans for long-term responsibility and stewardship

Concern 1 - regarding point (1) and (2)

The authors write:

"At the time of excavation, the people living around Kulubnarti did not identify remains in Christian-style graves as their ancestors and communicated this to Dr. Van Gerven and his colleagues"

This is problematic in regard to point (1). This means that perhaps the wrong people were asked for permission to study the site. Let me give a (perhaps not so) hypothetical example.

A set of researches want to study a Native American site in the United states. The original people who

lived there had been replaced by outsiders of European descent. The researchers ask for permission to study the grave sites from the current European occupants and they respond: 'We do not identify the remains in the Native American style graves as our ancestors'. Did the researchers in this example ask the right people for permission? Did they get meaningful consent to study the site?

What if a representative from the US government gave permission to study this site without asking the Native American community? Does this count as meaningful consent to study the site?

Related to point (2) - given the tension between the christian minority and Muslim majority in Sudan how do the authors plan on addressing this issue?

As the ancient individuals in question were christian, how did the authors engage with the present-day christian minority community in Sudan?

Concern 2 - regarding point (3)

The authors, when asked about how to communicate the findings to the entities of the studied communities, reply:

"The first author has made arrangements to carry out a dedicated return of results to both the academic community and to the public in Sudan. If it is safe and responsible to travel ..."

What happens if it is not safe to travel (very likely given the current pandemic)? What is the plan then for engaging with the community? Do the authors have established contacts within the community to allow the discussion of alternative strategies for discussing the results.

"The first author will discuss the results (does this mean a talk/workshop?) with staff and students at the University of Khartoum, where there are students studying archaeology and bioanthropology."

I noticed that there is not a single co-author with an affiliation to a university outside of Europe and the United States. What did the authors do to involve local researchers in this study to address point (3)? In addition, if they had followed the guidelines this would have potentially allowed one of these staff members or students to disseminate the results. Would a potential alternative be to fund the travel of a selection of these staff/students to Harvard to discuss with co-authors and then bring information back to the community.

Reviewer #4 (Remarks to the Author):

The authors have addressed my comments/suggestions.

As a final note, since the authors use Outlier analysis in their Bayesian modelling, I suggest they no longer note agreement indices. In terms of outlier detection, they should not be used (or at least

reported) in tandem. Agreement indices are a diagnostic tool meant to determine how well a posterior distribution agrees with the prior distribution. If outlier analysis is applied, you impose, a priori, that all dates are outliers to a specified probability. Therefore, if outlier analysis is used, one should guide the analysis based on this output. Although this is not explicitly stated by Bronk Ramsey in the 2009 publication, he does mention that one or the other should be used.

REVIEWER COMMENTS

Reviewer #1 (Remarks to the Author):

The revised article "Social stratification without genetic differentiation at the site of Kulubnarti in Christian Period Nubia" by Sirak et al. has covered precisely all the comments raised by the reviewers. I would like to thank the authors for their clear responses and for addressing the answers in the revised version.

I am glad to see that an Ethics statement was added to the results section. This statement weighs a lot on this well-written article and its novel data.

I would like to thank the authors for their responses to some comments that were probably raised because of unclarity in the original submission. In the revised version they have very well-considered and addressed them to help their readers avoid confusion. This will, without a doubt, attract the readers to the novelty of the data and the analysis conducted to provide pieces of evidence.

I am optimistic that the step taken by the authors to share their results with the communities and/or community representatives will allow aDNA studies to receive appreciation from non-scientific communities and the public.

We thank the reviewer for these positive comments. We are particularly grateful that they feel that the responses we provided in response to the requests for clarification about our ethical approach have addressed their questions.

Reviewer #2 (Remarks to the Author):

The authors have now updated their analysis and I have no further comments for the scientific analysis.

However based on the response to reviewers 1's question regarding the guidelines from the American Society of Human Genetics (ASHG) I have other concerns! The main points listed in Wagner et al 2020 are:

- (1) formally consult with communities
- (2) address cultural and ethical considerations
- (3) engage communities and support capacity building
- (4) develop plans to report results and manage data
- (5) develop plans for long-term responsibility and stewardship

Concern 1 - regarding point (1) and (2)

The authors write:

"At the time of excavation, the people living around Kulubnarti did not identify remains in Christian-style graves as their ancestors and communicated this to Dr. Van Gerven and his colleagues"

This is problematic in regard to point (1). This means that perhaps the wrong people were asked for permission to study the site. Let me give a (perhaps not so) hypothetical example.

A set of researches want to study a Native American site in the United states. The original people who lived there had been replaced by outsiders of European descent. The researchers ask for permission to study the grave sites from the current European occupants and they respond: 'We do not identify the remains in the Native American style graves as our ancestors'. Did the researches in this example ask the right people for permission? Did they get meaningful consent to study the site? What if a representative from the US government gave permission to study this site without asking the Native American community? Does this count as meaningful consent to study the site?

Wagner et al.'s recommendations are a valuable contribution to the discussion about ethical approaches to ancient DNA studies, but in fact they come very much from the perspective of engagement with Indigenous peoples in the United States (US). As evidence for this, all the co-authors of Wagner et al. are US-based; their research largely revolves on Indigenous North Americans; and the basis for ethics that they build on is US-based legislation and the discourse that has developed around it (the North American Graves Protection and Repatriation Act – NAGPRA). Thus, an important question is whether their five recommendations apply worldwide, including to contexts like studies of people living in Christian Period Sudan.

To explore the extent to which the Wagner et al. recommendations are generalizable worldwide, and to build on them, the first and last authors of the present-study co-led the organization of a virtual workshop on ethics in ancient DNA, which happened in November 2020, and involved more than 60 scholars from more than 20 countries. The participants included museum curators, Indigenous people from the Americas, Australia, and Sub-Saharan Africa (including Sudan), and archaeologists and anthropologists who have engaged in some way with ancient DNA (geneticists were a minority in attendance). An explicit goal was to build on the Wagner et al. study, exploring the extent to which the recommendations were generalizable and the extent to which alternative approaches need to be taken to develop ethical approaches that work worldwide.

The result of our workshop was a set of five revised recommendations that we believe are globally applicable; the manuscript elaborating on these principles is in submission for publication. **[editor's note: this is now published at <https://doi.org/10.1038/s41586-021-04008-x>].** These are the following, and below we describe how we have abided by each in the present study:

(1) Researchers must abide by all regulations in the places where they work and from which their samples were drawn. Our manuscript describes the formal permissions we obtained for the export and analysis of the Kulubnarti Nubians.

(2) Researchers must prepare a detailed plan prior to beginning any study. This work which represents the Ph.D. thesis of first author Kendra Sirak was funded by the National Science Foundation (Grant No. BCS-1613577), and in this context requiring the submission of a detailed research plan.

(3) Researchers must minimize damage to human remains. Analysis of the Kulubnarti skeletal collection involved preferentially selecting individuals with accessible petrous bones that were disarticulated from the rest of the skull to maintain as many complete skulls as possible for any future analyses.

(4) Researchers must ensure that data are made available following publication to allow critical reexamination of scientific findings. The paper commits to place all aligned genomic sequences will be deposited upon publication into the European Nucleotide Archive (accession number PRJEB42975).

(5) Researchers must engage with stakeholders and ensure respect and sensitivity to stakeholder perspectives. The members of the community who lived in the village neighboring the Kulubnarti cemeteries and who used the mainland cemetery for the burial of their own deceased consented to the research following conversations with Dr. Dennis Van Gerven (co-senior author on this manuscript) before and during excavation. Reviewer #2 suggests that modern Christians in Sudan might be the appropriate stakeholder group for early Christians rather than the current resident population of Kulb and asks us to consider consulting them before the paper is published. However, a connection between modern Christians and ancient Christians in Sudan is not supported by the evidence: the present-day Christian community in Sudan lives in a different region and likely came from elsewhere. After extensive fieldwork and study in the region, we are confident that from a cultural historical perspective there are no living Nubians more closely linked to the Christians we excavated than the current residents of Kulb who willingly permitted and participated in the excavation.

We appreciate Reviewer #2's concern that "perhaps the wrong people were asked for permission to study the site," especially in light of the Wagner et al. recommendation that an ancient DNA study should "formally consult with communities." However, we did extensively consult with the local community living in Kulb as we describe in our Ethics Statement, and we do not believe it is the place of the researcher to tell any community who self-identifies as stakeholders that they are the "wrong" people to be considered based on differences in religious beliefs. A retroactive determination that the geographically proximate community which happens to be of the Muslim faith represents the "wrong" people to have consulted about this research may be harmful to the people presently living in this area who are plausibly descendants.

Speaking more generally, not only does our study fully conform to the strong ethical principles of our November 2020 workshop and the manuscript that we wrote following it that is currently under review, but it is also the case that scholars in the workshop who are specialists in African anthropology and archaeology and include among them African scholars from Sudan and Kenya) felt that applying some of the Wagner et al. principles could be actively harmful in an African context. Here is the relevant paragraph from our manuscript in review, which was co-written by Africanists (Wagner et al. did not include any authors who were experts on African archaeology or anthropology or African themselves):

"The meaning of Indigeneity varies globally. In Africa, descendants of colonized groups are now overwhelmingly in power, and Indigeneity often refers more to political or social marginalization on the basis of identity than to

traditions of how long groups have been established in a region. Many African communities have complex connections to the lands on which they live, including histories of colonial and postcolonial displacement and disruption. In some regions, people do not recognize past local populations as their relatives; this may be due to contemporary religious or cultural belief systems being different from past ones, records of migrations from elsewhere, fear of reprisal for being linked with other groups, and the continuing aftershocks of decisions made during European colonization that fractured socio-political landscapes and still contribute to violence and displacements... In such cases, centering Indigeneity as a principle for permitting ancient DNA analysis would likely be harmful.”

In summary, we appreciate the reviewer raising these issues and can understand how the recent recommendations on these topics have provoked greater consideration of whether ancient DNA work was carried out ethically. However, in light of the explicit and extensive discussions we have had about these topics—prompted precisely by our own concerns about the best way to proceed which we share with the reviewer—and the guidance on this we have had from African scholars including from Sudan, we think the suggestion to carry out additional consultation for this study with communities living potentially hundreds of kilometers away from Kulb has the potential to be harmful and thus unethical. It has the potential to second-guess the local community we have already consulted with and that views itself as the appropriate group with which to consult, to potentially undermine their claim to speak for their past, and thereby to promote conflicts between groups that were not in conflict before.

Related to point (2) - given the tension between the christian minority and Muslim majority in Sudan how do the authors plan on addressing this issue?

As the ancient individuals in question were christian, how did the authors engage with the present-day christian minority community in Sudan?

After consultation with African scholars, we have deep concern that seeking out Christians who live hundreds of kilometers away from Kulb and treating them as stakeholders for the archaeological work at Kulb who have more right to speak for the ancient people who lived there than the local (Muslim) community could contribute to tension (see above for our discussion of this). This is an example of how the Wagner et al. recommendations are not appropriate everywhere, and could lead to unethical outcomes in some contexts (by promoting conflict and tension if followed to the letter in contexts for which they were not designed – that is, contexts not well described by a history of settler colonialism that characterizes the US).

This does not mean that modern Christian communities should under no circumstances be seen as stakeholders for the early Christians who lived at Kulubnarti. If there were modern Christian communities that had traditions of descent from groups like those who lived at Kulb and who at some point in history left the region either due to being unwillingly pushed out or for other reasons, they would be plausible stakeholders in addition to the local community we consulted. However, there is no evidence that this is the case here. As we show in our work, the individuals from Kulubnarti that we study are not the direct ancestors of present-day genotyped people from Nubia following the Christian Period, and the authors of our paper who include experts on Nubian anthropology and history are not aware of any modern Sudanese Christian communities that see themselves as connected specifically to the early Christian communities who lived in places like Kulb. As such, it would not be appropriate to attempt to identify a present-day Sudanese Christian community to engage simply based on the sharing of a common religion with the people from Kulubnarti that we study in this work. The majority of Sudanese Christians today who live in the northern part of the country which is where Kulubnarti is found (and who comprise only ~1% of the population) are mostly descended from groups who moved to Sudan in the recent centuries, making it nearly impossible to identify who may be a stakeholder without challenging individual notions of identity and belonging in a way that can possibly have dangerous repercussions for these individuals.

We agree with the importance of clarifying this issue, however, and in our revision we have added a sentence to the Ethics Statement explaining why we treated the current population of Kulb as the appropriate group with which to engage to obtain community perspectives on the research.

Our Ethics Statement now reads:

“We acknowledge the ancient individuals whose remains we analyzed in this work and who must be treated with respect. Excavation of human remains from the two Kulubnarti cemeteries occurred in 1979 under a license granted by the Sudan Antiquities Service (now the National Corporation for Antiquities and Museums) to Dr. William Y. Adams; the excavation of the cemeteries was funded by the National Science Foundation (Grant No. 77-270210-535),

and led by Dr. Dennis Van Gerven, co-senior author on this work; it was undertaken as a part of the UNESCO International Campaign to Save the Monuments of Nubia. Prior to the excavation of the Kulubnarti cemeteries, the head of the Sudanese Antiquities Service approved the research plan, including the invasive investigations (such as biochemical analyses) anticipated at the time. All graves to be excavated were marked by the archaeological team and their excavation was approved by the Nubian reiss (foreman). The excavation was inspected by a representative of the Sudan Antiquities Service monthly, and all excavated remains were reinspected prior to their export, which occurred in accordance with the regulations at that time. At the time of excavation, although the people living around Kulubnarti did not identify remains in Christian-style graves as their ancestors and communicated this to Dr. Van Gerven and his colleagues, they lived in close geographic proximity to the site and continued to use the R cemetery as a burial ground for their deceased. Based on this connection, they provided community perspectives as the custodians of the cemetery and gave their consent for this work. Following the wishes of the local community, Dr. Van Gerven's team avoided any interference with Muslim graves in the Kulubnarti cemeteries, which were distinguishable by their north-south orientation."

Concern 2 - regarding point (3)

The authors, when asked about how to communicate the findings to the entities of the studied communities, reply:

"The first author has made arrangements to carry out a dedicated return of results to both the academic community and to the public in Sudan. If it is safe and responsible to travel ..."

What happens if it is not safe to travel (very likely given the current pandemic)? What is the plan then for engaging with the community? Do the authors have established contacts within the community to allow the discussion of alternative strategies for discussing the results.

The first author (Sirak) is presently a team member on two active research projects in Sudan that also involve personnel from NCAM and the University of Khartoum, allowing her to establish the necessary connections to carry out a dedicated return of results from the Kulubnarti project that is effective in reaching both the academic and nonacademic communities in Sudan. While the ancient DNA data from Kulubnarti represents a valuable step toward revealing the ancient genetic landscape of Sudanese Nubia, the first author has now established a network of collaborators who are from and who work in Sudan and who will be involved in upcoming research projects. If it is impossible to travel to Sudan in November/December 2021 as is currently planned, the trip will be postponed until the following year. This trip will still represent meaningful and serious engagement and contribution to capacity building, whether it occurs this year or next.

"The first author will discuss the results (does this mean a talk/workshop?) with staff and students at the University of Khartoum, where there are students studying archaeology and bioanthropology."

Our commitment for communication of results to Sudanese communities—already planned and written into active research proposals—include:

- 1. The first author will give a lecture and a day-long workshop on the "What's, how's, and why's of ancient DNA research" that will include a presentation of the Kulubnarti results to students of archaeology and anthropology at the University of Khartoum;*
- 2. The first author will give a lecture and a half-day-long workshop on the "What's, how's, and why's of ancient DNA research" that will include a presentation of the Kulubnarti results to personnel at NCAM;*
- 3. The first author will give a talk to the general public about ancient DNA research and hold a discussion session where questions and concerns about the research can be voiced; we are planning to hold this event at the British Council in Khartoum (venue to be confirmed this summer);*
- 4. The first author will provide training to collaborator Dr. Mohamed Bashir (University of Khartoum) and his colleagues on the methods for sampling skeletal material for ancient DNA research; this capacity building activity comprises an important part of their collaborative National Geographic project. Training will how to carry out minimally-invasive method for sampling intact skulls that the first author published in Sirak et al. 2017. The first author will provide Dr. Bashir with the necessary equipment to carry out sampling*

independently in the future, ensuring that our collaborative work also generates new opportunities for him and his colleagues to expand their role in ancient DNA research in future projects.

5. *The first author will create a poster detailing the results from the Kulubnarti project that will be shared with NCAM personnel for placement in the Sudan National Museum.*

I noticed that there is not a single co-author with an affiliation to a university outside of Europe and the United States. What did the authors do to involve local researchers in this study to address point (3)? In addition, if they had followed the guidelines this would have potentially allowed one of these staff members or students to disseminate the results. Would a potential alternative be to fund the travel of a selection of these staff/students to Harvard to discuss with co-authors and then bring information back to the community.

We fully agree with the reviewer's view that is important to include local researchers in scholarly output whenever possible and contribute to capacity building initiatives. This research occurred as part of a Ph.D. project on a collection of human remains that were already curated in the US, however, and because there was not a component of this part of research that occurred in Sudan, it did not involve Sudanese researchers as collaborators. To add in Sudanese co-authors following the research simply for the optics of including them on scholarly output would be dishonest, as authors should only be included in a paper if they contributed directly to the generation and analysis and interpretation of data. More generally, we note that the excavations that produced the anthropological collection analyzed in this study occurred about four decades ago, and while they did involve serious engagement as described in our Ethics Statement, it was not with local scholars (consistent with the standards of the time), and we cannot retroactively change this. Instead we must proceed from the point we are at and try to do things in the most ethically committed way possible going forward, which we have done here.

As pointed out by the reviewer, the involvement of local researchers is important. To improve the inclusion of Sudanese colleagues in future work, the author has already committed to carry out a return of results to Sudan (see the previous section), and will provide the training in skills that enables Sudanese researchers to meaningfully participate as co-authors in upcoming research.

Reviewer #4 (Remarks to the Author):

The authors have addressed my comments/suggestions.

As a final note, since the authors use Outlier analysis in their Bayesian modelling, I suggest they no longer note agreement indices. In terms of outlier detection, they should not be used (or at least reported) in tandem. Agreement indices are a diagnostic tool meant to determine how well a posterior distribution agrees with the prior distribution. If outlier analysis is applied, you impose, a priori, that all dates are outliers to a specified probability. Therefore, if outlier analysis is used, one should guide the analysis based on this output. Although this is not explicitly stated by Bronk Ramsey in the 2009 publication, he does mention that one or the other should be used.

We have made the change recommended by this reviewer (excluding agreement indices for individual dates); however, we retained the Agreement Index for the model as a whole (AModel), as it is customary to include it.

Reb BREVIEWER COMMENTS

Reviewer #2 (Remarks to the Author):

The reviewers have addressed my concerns and it is now clear to me that they have thought a lot about the ethic implications of this study.

Reading the previous version of the manuscript and supplement it was difficult to know to which extend the community was engaged in this study and exactly what the plan was to communicate the findings.

I don't think the authors need to add anything to the manuscript but instead they can refer to the online "response to reviewers" in case future readers want to know which ethical concerns were considered or what the concrete plan for engaging with the local community is.

I have no further comments and believe this manuscript is now suitable for publication.

Reviewer #5 (Remarks to the Author):

The manuscript by Sirak .et al., is a valuable contribution to genetic anthropology and the field of ancient DNA analysis increasing the number of ancient individuals with genome-level data from the Nile Valley from three to 69 as claimed by authors. Of significance is that this addition takes place in a geographical area that bears exceptional importance to human history and evolution, yet sadly is amongst the most understudied globally. Although the authors presented some spectacular analyses of remains in parts of Nubia in northern Sudan, the conclusions are marred by major weaknesses in the interpretation of data that may undermine their conclusions. Following are few comments that attempt to bring to attention of the authors some of these concerns:

General comments

I would first like to pinpoint some conceptual conjectures that appeared in the introduction. The intention is to encourage a critical view of the literature on Sudan rather than repeating historical clichés like the one referring to Nubia as corridor, a concept which is borrowed from archeology and features in the title of Adams famous volume on Nubia. This notion builds on historical narratives that undermine the role and cultural contributions of these parts of Africa reducing it to that of a geographical passage for the movement of goods and populations. With more recent knowledge, one could hardly name a location that has been the scene of such pivotal human cultural evolution as merely a corridor.

Another tenet in the analysis is the historical notion of admixture that follows on the previous dictum on demographic movements. Admixture between populations of Sudan with Asian groups is one non-substantiated beliefs that lacks support from current analysis of allele frequencies in modern Sudanese nor by knowledge of large demographic events or back migrations prior to 19th century occupation of northern Sudan.

I quote the authors "Genetic studies of present-day Nubians reveal a mix of sub-Saharan African- and

West Eurasian-related ancestry, but the mixture is largely a result of the Arab conquest of the late-1st and early-2nd millennia.”

Despite the inherent problems of terms like sub-Saharan Africa and its relevance to earlier demography in the region and the history of the Sahara itself, Arab conquest is known almost by every single Sudanese to have been routed at the doors of Nubia, resulting in what is known as the "pact" treaty. The subsequent slow infiltrations of nomads from Western African Sahara or Arabia settling gradually at the fringes of agricultural societies in communities that value land ownership as social status is evident in the Y chromosome analysis of the current Sudanese gene pool. In fact the last Christian stronghold fell to a pact of pagans and Arab speaking pastoralist as recently as early sixteenth century.

Both Authors and reviewers made an issue out of religion of Sudanese in tally with a public image formed by the international media on Sudan reflecting stark lack of knowledge on the question of Culture and religion in the country. Nubians of today are some of the last to convert to Islam and abundant Christian and pre Christian traditions still exists. They embrace with pride and no reservation these legacies. Rather than religion it is perhaps demographic reasons that are behind what is stated by the authors: "At the time of excavation, the people living around Kulubnarti did not identify remains in Christian-style graves as their ancestors and communicated this to Dr. Van Gerven and his colleagues" And is indeed expected in an area with continuous flux of populations. Preliminary analysis of ancient DNA from Northern Sudan indicated such discontinuities (Hisham Hassan, unpublished) And as pointed by authors as well: "Present-day Nubians are not directly descended from the Christian Period people from Kulubnarti without additional admixture, attesting to the dynamic history of interaction that continues to shape the cultural and genetic landscape of Nubia."

Population structure:

It is obvious from the robust and extensive analysis availed by genome wide data for these cemeteries that the Kulubnarti population analyzed is of limited effective size attested by ROH and other genetic markers. Tracts of homozygosity are not common either in Africa or among healthy Sudanese populations due to the large east African effective population size and the peculiar patterns of LD. However the author's central theme in the manuscript is based on lack of genetic differentiation of the settlers despite indications of social differences. One cemetery is on an island while the other is on the west bank of the Nile or a historical branch of it. This type of habitation and settlement is still the case along the Nile with Island dwellers usually being earlier settlers and of relatively high social status, which does not seem to be the case here. Social status is not a direct indication of class and does not necessarily correspond with economic social stratification in a society that has remained egalitarian in nature until recently. Socio economic data are not presented in the manuscript to reach such clear-cut conclusions specially ones that feature so prominently in the title of the manuscript. The authors rather base their argument on evidence of health disparities between the two cemeteries. The notion of diseases afflicting low social stratus more is modern. There are multiple alternative explanations including consanguinity. A small population size is more likely to exhibit a spectrum of diseases including autosomal recessive.

Ancestry analysis

Authors carried out ancestry analysis using mitochondrial DNA and Y chromosome. The results are contradictory to their main theme of admixture between east Africans (Nilotics) and the Levant. Such

admixture is not substantiated by bona fide markers of Nilotic and Levant descent. Neither Group A of the Nilotics nor J2 are seen as far as males are concerned. Authors rather claim a major Levant contribution to the Kulubnarti female gene pool. This is contradictory also to published literature on female expansion/migrations studied in published data and data from Sudan that shows the female gene pool to be the outcome of in situ evolution (Osman et al., 2021) where some of the so called out of Africa haplotypes are found to be present in substantial frequencies in samples from populations not known to have left the continent and that display fixation of genetic markers of continuous population history in the continent (Albasheer et al., 2020), in addition of course to historical and archeological evidence. Such dearth of analysis of native mtDNA might explain misnamed and faulty classifications of human uniparental haplogroup assignments and hence the ensuing faults around dates and patterns of migration. Unfortunately migration patterns are still based on historical accepted narratives and lack substantiated solid proof.

For example haplogroup H, used to justify major female gene flow in this manuscript (irrespective of indicators like culture or language), was shown to exist among Hausa and Fulani, two groups that shared recent history in the Sahel (Osman et al., 2021). Nubian influx itself is being proposed by archeologists to have begun from west of the Nile at some time three millennia ago.

Assigning populations

Several terms were used to assign populations. Using the term Nilotic to denote Nilo-Saharan populations is a mistake, as Nilotics has been confined to a branch of Nilo-Saharan speakers mainly in southern Sudan, Ethiopia and the great lakes. Similarly is the term Sudanese Arab. Arabic became the lingua franca in Sudan for at least a century for some time, the term Arab is uninformative of Ethnicity if not deceiving. Even "Arabic speaking" which is descriptive of cultural communities should include the name of the people/ethnic group.

Ethics and community engagement:

The authors utilize a rather outdated approval and consent that dates back to the "Campaign to Save the Monuments of Nubia". "Prior to the excavation of the Kulubnarti cemeteries, the head of the Sudanese Antiquities Service approved at the time ", something that is not compatible with present day good practices in ethics and research which include feeding back results to community, stewardship etc.... individuals' Perception of research aspects of identity and history can be quite dynamic. Also community engagement should not be confined to those isolated vulnerable communities in the site where research and excavation is carried out. Broader consent and consultation, which should extend to include the natural stewards and intellectual guardians of the heritage of these communities, such as local academic and educational institutions, no matter how poor their technological setting or academic tradition. This is not essential only for "political correctness" but for reasons that pertain, in addition to ethics, to the scientific rigor of the content, the representativeness of historical narratives and validity of conclusions in this case. Genuine and actual contributions should be the criteria leading to authorship or acknowledgement for both local and international scientists alike rather than token-like "piggybacking" attributions.

References

Maha M. Osman, Hisham Y. Hassan a,1, Mohammed A. Elnour a,1, Heeran Makkan c, Eyoab Iyasu Gebremeskel a,d, Thoyba Gais a, Mahmoud E. Koko a, Himla Soodyall c, Muntaser E. Ibrahim a,*

Mitochondrial HVRI and whole mitogenome sequence variations portray similar scenarios on the genetic structure and ancestry of northeast Africans. *Meta Gene* 27 (2021) 100837

Musab M. Ali Albsheer, Ayman Hussien, Dominic Kwiatkowskic, Muzamil Mahdi Abdel Hamida, Muntaser E. Ibrahima. The Duffy T-33C is an insightful marker of human history and admixture *Meta Gene* 26 (2020) 100782

).

REVIEWER COMMENTS

Reviewer #2 (Remarks to the Author):

The reviewers have addressed my concerns and it is now clear to me that they have thought a lot about the ethic implications of this study.

Reading the previous version of the manuscript and supplement it was difficult to know to which extend the community was engaged in this study and exactly what the plan was to communicate the findings.

I don't think the authors need to add anything to the manuscript but instead they can refer to the online "response to reviewers" in case future readers want to know which ethical concerns were considered or what the concrete plan for engaging with the local community is.

We are happy with the suggestion of making the reviews and our responses to them fully available with publication through the Transparent Peer Review mechanism. We agree that this will better help readers understand the ethical issues that arise in a study like this and how we addressed them.

I have no further comments and believe this manuscript is now suitable for publication.

Reviewer #5 (Remarks to the Author):

The manuscript by Sirak .et al., is a valuable contribution to genetic anthropology and the field of ancient DNA analysis increasing the number of ancient individuals with genome-level data from the Nile Valley from three to 69 as claimed by authors. Of significance is that this addition takes place in a geographical area that bears exceptional importance to human history and evolution, yet sadly is amongst the most understudied globally. Although the authors presented some spectacular analyses of remains in parts of Nubia in northern Sudan, the conclusions are marred by major weaknesses in the interpretation of data that may undermine their conclusions. Following are few comments that attempt to bring to attention of the authors some of these concerns:

General comments

I would first like to pinpoint some conceptual conjectures that appeared in the introduction. The intention is to encourage a critical view of the literature on Sudan rather than repeating historical clichés like the one referring to Nubia as corridor, a concept which is borrowed from archeology and features in the title of Adams famous volume on Nubia. This notion builds on historical narratives that undermine the role and cultural contributions of these parts of Africa reducing it to that of a geographical passage for the movement of goods and populations. With more recent knowledge, one could hardly name a location that has been the scene of such pivotal human cultural evolution as merely a corridor.

We had not previously heard concerns about the use of the term "corridor" in reference to Nubia. However, after reading the referee's reply, we fully appreciate how this term could make people living in this region feel that the importance of the region is minimized, as it could be read as a suggestion that Nubia is simply a passage between one place and another and not a place that was a primary region where goods, cultures, and genes were exchanged in a way that contributed to the development of a dynamic culture and diverse population. This kind of response is exactly why it is so valuable to have a review from a Sudan-based

scholar, and we appreciate the opportunity to take on board their perspectives in our revision to ensure that our final perspective is more sensitive to local perspectives.

In response to this referee's comment, we have adjusted our language and edited the manuscript to remove the word "corridor." Our manuscript now reads: "Nubia has been a primary center for the meeting and mixing of people from sub-Saharan Africa, Egypt, and West Eurasia since prehistory..." and "Situated along the Nile River between the First Cataract at Aswan in present-day Egypt and the confluence of the Blue and White Nile Rivers near the present-day Sudanese capital of Khartoum (Fig. 1a), Nubia has seen a long and dynamic history of continual human occupation and occupied a central position as a place where people from multiple parts of Africa and West Eurasia lived and interacted¹⁻⁵."

Another tenet in the analysis is the historical notion of admixture that follows on the previous dictum on demographic movements. Admixture between populations of Sudan with Asian groups is one non-substantiated beliefs that lacks support from current analysis of allele frequencies in modern Sudanese nor by knowledge of large demographic events or back migrations prior to 19th century occupation of northern Sudan.

In this manuscript, we carefully use the term "West Eurasian-related" to refer to ancestry that is related to that present in West Eurasia but was plausibly also present for very long periods of times in parts of northeastern Africa (however, we do not have very old DNA from northeastern Africa, so the only actual ancient DNA reference point we can use is that from West Eurasia). To address this referee's comment and to avoid the potential of future confusion, we have further clarified what we mean by "West Eurasian-related" in this context, editing our revised manuscript to include the statement: "although we refer to this ancestry here as "West Eurasian-related" because we do not yet have ancient genetic data from an appropriate phylogenetically-adjacent reference group from Africa which is likely to have been its proximate source."

I quote the authors "Genetic studies of present-day Nubians reveal a mix of sub-Saharan African- and West Eurasian-related ancestry, but the mixture is largely a result of the Arab conquest of the late-1st and early-2nd millennia." Despite the inherent problems of terms like sub-Saharan Africa and its relevance to earlier demography in the region and the history of the Sahara itself, Arab conquest is known almost by every single Sudanese to have been routed at the doors of Nubia, resulting in what is known as the "pact" treaty.

The terms "sub-Saharan African-related" and "West Eurasian-related" as descriptors of ancestry are emphatically not statements about geographic origin. Rather, by adding the qualifier "-related" following the use of any ancestry term (for example, we always write "sub-Saharan African-related"), we are carefully stating that the ancestry type we are discussing is related to that deriving from a particular geographic area but not necessarily from that region. Lopez et al. Nature Communications 2021 utilize the term "Egyptian-related" in the same sense. In response to the referee's comment and to clarify this further, we have added the following parenthetical statement to our revised manuscript: "in this work we use the qualifier "-related" when the ancestry we are discussing is related to that deriving from a particular geographic area but not is necessarily from that region itself."

The referee is absolutely right to raise the important historical point that the Arab conquest halted for at least some time in Nubia and that the Baqt treaty played an important role in determining the course of Nubian-Muslim migrations for many centuries. Nevertheless, many lines of evidence support the substantial influence Arab immigration had on Nubia not just culturally but also demographically. Egypt came under the political control of Arabs originating from the Arabian Peninsula and practicing Islam beginning in 639 C.E. and culminating with the surrender of Alexandria in 641 C.E. (Adams 1977; Shinnie 1996). The

Arab armies then penetrated into Christian Nubia as far south as Dongola but were met with Nubian resistance and eventually defeated (Hasan 1967). A second invasion into Nubia was mounted in 651–652 C.E., during which the Muslim army reached as far up the Nile as Old Dongola, but this campaign ended inconclusively and resulted in the Baqt treaty truce (Shinnie 1996). The Baqt treaty contributed to prosperity in Nubia by institutionalizing trade relations between the Nubians and the Islamic world, a process that also “led in the long term to a prolonged presence of Arab merchants in Nubia” (Jakobielski 1987: 232). This treaty therefore provided an opportunity for contact and gene flow between resident Nubians and Muslims, and by comparing ancient genetic data such as we report here to modern genetic data from Nubia we can in principle quantify the degree of these demographic impacts. Our genetic analysis as well as that of others suggest that Arab contacts did eventually have a major impact on Nubia. For example, Hollfelder et al. 2017 PLoS Genetics report that “We estimate the admixture in current-day Sudanese Arab populations to about 700 years ago, coinciding with the fall of Dongola in 1315/1316 AD, a wave of admixture that reached the Darfurian/Kordofanian populations some 400±200 years ago.” We acknowledge and cite this in our manuscript.

The subsequent slow infiltrations of nomads from Western African Sahara or Arabia settling gradually at the fringes of agricultural societies in communities that value land ownership as social status is evident in the Y chromosome analysis of the current Sudanese gene pool. In fact the last Christian stronghold fell to a pact of pagans and Arab speaking pastoralist as recently as early sixteenth century.

Both Authors and reviewers made an issue out of religion of Sudanese in tally with a public image formed by the international media on Sudan reflecting stark lack of knowledge on the question of Culture and religion in the country.

We respect the referee’s perspective here. However, we continue to believe it is relevant to our study that the individuals we are analyzing have Christian-style burials. This is a distinctive aspect of the culture of this time that influenced many aspects of life, and it is a legitimate question to try to explore what the genetic correlates of people associated with this culture were. Indeed, our finding of an appreciable rate of 8-20cM stretches of homozygosity confirms that marriage patterns were somewhat distinctive from populations living in the region today. As we write: “The finding that intermediate ROH (8-20cM) was more common points toward Kulubnarti functioning as a small community that mostly mated among themselves but also exchanged mates with a bigger meta-population...”

Nubians of today are some of the last to convert to Islam and abundant Christian and pre Christian traditions still exists. They embrace with pride and no reservation these legacies. Rather than religion it is perhaps demographic reasons that are behind what is stated by the authors: "At the time of excavation, the people living around Kulubnarti did not identify remains in Christian-style graves as their ancestors and communicated this to Dr. Van Gerven and his colleagues"

And is indeed expected in an area with continuous flux of populations. Preliminary analysis of ancient DNA from Northern Sudan indicated such discontinuities (Hisham Hassan, unpublished) And as pointed by authors as well: "Present-day Nubians are not directly descended from the Christian Period people from Kulubnarti without additional admixture, attesting to the dynamic history of interaction that continues to shape the cultural and genetic landscape of Nubia."

We accept and appreciate the referee’s statement that many modern Sudanese living in overwhelmingly Muslim regions today such as the one around Kulubnarti may feel a strong connection to pre-Muslim populations. That said, the research team at the time of excavation did intentionally engage with the local people from Kulubnarti about their concerns and sensitivities. As we describe in our paper, these local

community members communicated to co-senior author Van Gerven that they did not identify remains in Christian-style graves as their ancestors.

Population structure:

It is obvious from the robust and extensive analysis availed by genome wide data for these cemeteries that the Kulubnarti population analyzed is of limited effective size attested by ROH and other genetic markers. Tracts of homozygosity are not common either in Africa or among healthy Sudanese populations due to the large east African effective population size and the peculiar patterns of LD. However the author's central theme in the manuscript is based on lack of genetic differentiation of the settlers despite indications of social differences. One cemetery is on an island while the other is on the west bank of the Nile or a historical branch of it. This type of habitation and settlement is still the case along the Nile with Island dwellers usually being earlier settlers and of relatively high social status, which does not seem to be the case here. Social status is not a direct indication of class and does not necessarily correspond with economic social stratification in a society that has remained egalitarian in nature until recently. Socio economic data are not presented in the manuscript to reach such clear-cut conclusions specially ones that feature so prominently in the title of the manuscript. The authors rather base their argument on evidence of health disparities between the two cemeteries. The notion of diseases afflicting low social stratus more is modern. There are multiple alternative explanations including consanguinity. A small population size is more likely to exhibit a spectrum of diseases including autosomal recessive.

We absolutely agree with the referee that social status is not necessarily related in a simple way to class – this is a very important point that always needs to be kept in mind. Instead, the question we attempt to address is whether the systematic differences that have previously been observed between the two cemeteries based on physical anthropology, which have been hypothesized to be possibly related to wealth or class might have a connection to family grouping or ancestry which could provide a new clue to the source of the differences that are observed. The negative answer we obtain, that is, the fact that we find no evidence for a systematic difference between the ancestries of people buried in the two cemeteries, is an interesting finding which informs our understanding when jointly considered with the physical anthropology data that does find a difference between the cemeteries.

Specifically, the entire focus of this manuscript is to utilize a new type of data to test an existing hypothesis: that Kulubnarti was home to a culturally and genetically homogenous population that was divided into two groups that lived separately and utilized separate burial grounds, and that these groups were possibly socially-stratified. We rely on decades of archaeological and bioarchaeological research indicating that people buried in the S cemetery experienced more stress and disease and died younger than those buried in the R cemetery and do not attempt to evaluate the socio-economic standing of these two cemetery groups, but instead to determine whether they are consistent with forming a single genetic population. We are cautious to note that the previous archaeological, bioarchaeological, and ethnographic evidence suggests plausible social stratification at the site, writing: “Driven by questions inspired by archaeology and bioarchaeology, our analysis provides new insight into the ancestry of Christian Period people from Kulubnarti and into the genetic relationships among individuals buried in two cemeteries with significant differences in morbidity and mortality suggestive of social stratification.”

We take care not to explicitly attribute our results to socio-economic differences or non-differences. This can be seen in the way we describe our results – writing, for example, “Here, ancient DNA provides a new line of evidence supporting the hypothesis that the burial of people in two cemeteries at Kulubnarti was not strongly rooted in genetic differences. Instead, this burial pattern may reflect social or socioeconomic differences that are not yet fully known, or may be a cultural practice, such as the burial of unbaptized individuals or those suffering from particular illnesses, apart from the rest of the population” – note that

*we interpret our data only as supporting that there are no genetic differences among the groups and that we also note that the observed pattern of differences in experience that are skeletally-evident *may* reflect social or socioeconomic differences, or represent other practices (and we suggest some alternative possibilities, such as the burial of unbaptized individuals in a separate area).*

Ancestry analysis

Authors carried out ancestry analysis using mitochondrial DNA and Y chromosome. The results are contradictory to their main theme of admixture between east Africans (Nilotics) and the Levant.

Our genome-wide data and data from uniparental haplogroups are in fact fully consistent with evidence of admixture at Kulubnarti, and we highlight this. For example, we write: "...we find that 35 out of 63 individuals from both cemeteries who were not first-degree relatives sharing a maternal lineage belong to 11 mitochondrial DNA (mtDNA) haplogroups that are presently distributed predominantly in West Eurasia although the presence of such lineages for thousands of years in northeastern Africa as well has been clearly established by previous work^{31,64,65}. The observation of 35 individuals carrying mtDNA haplogroups that are most common in West Eurasia is what would be expected for anywhere between 43–68% of maternal ancestry at Kulubnarti coming from West Eurasian ancestors via northeastern Africa (based on evaluating whether each proportion in this range included 35 West Eurasian mitochondrial haplogroups within its 95% central CI), which overlaps the 59–77% estimate of West Eurasian-related ancestry deriving from females made by comparing ancestry proportions on the autosomes and X chromosome (Methods)." While we find that more males from the R cemetery belong to haplogroups on the Y chromosome E branch and most males from the S cemetery belonged to haplogroups that were likely West Eurasian in origin, we also show this difference is not statistically significant ($p=0.11$).

Such admixture is not substantiated by bona fide markers of Nilotic and Levant descent. Neither Group A of the Nilotics nor J2 are seen as far as males are concerned.

The reviewer is correct in noting the absence of Y chromosome haplogroups A or J2 at Kulubnarti. One possible reason for this, however, is limited sample size: we analyze a relatively low number of male individuals (28 unrelated males), the majority of whom belong to haplogroups on the E1b1b1 (E-M215) branch that is hypothesized to have originated in northeast Africa ~25 kya⁷⁰ and is commonly found in present-day Afro-Asiatic speaking group. That we find this haplogroup at such high prevalence is not surprising given what we know about its distribution in this part of northeastern Africa. The non-E haplogroups are sufficiently few given our limited sample size that the absence of Y chromosomes A or J2 is not statistically surprising. In this work, the key evidence of admixture is from genome-wide data, which effectively represents thousands of independent phylogenetic trees (instead of a single tree that is represented by a uniparental marker) and is therefore a much more powerful tool for reconstructions of population history, including admixture.

Authors rather claim a major Levant contribution to the Kulubnarti female gene pool. This is contradictory also to published literature on female expansion/migrations studied in published data and data from Sudan that shows the female gene pool to be the outcome of in situ evolution (Osman et al., 2021) where some of the so called out of Africa haplotypes are found to be present in substantial frequencies in samples from populations not known to have left the continent and that display fixation of genetic markers of continuous population history in the continent (Albasheer et al., 2020), in addition of course to historical and archeological evidence. Such dearth of analysis of native mtDNA might explain misnamed and faulty classifications of human uniparental haplogroup assignments and hence the ensuing faults around dates and patterns of migration. Unfortunately migration patterns are still based on historical accepted narratives and lack substantiated solid proof.

We value the findings of Osman et al. 2021, which analyzes the frequencies of mitochondrial DNA haplogroups in present-day Nubians without analyzing ancient DNA; we have now added the Osman et al. 2021 citation in our work. However, in this study, we are analyzing ancient DNA from Nubia and not modern DNA. As we show in our study, the present-day groups from Nubia for which we have genome-wide data are not consistent with being directly descended from people closely related to the ancient Kulubnarti Nubians, suggesting that studying the genetic patterns in these modern Sudanese may not provide reliable information about the ancient people. The only way we can reliably understand the ancestry of these ancient Nubians is through ancient DNA. In our revised manuscript we now add a sentence that highlights the Osman et al. 2021 work as well as the Holfelder et al. 2017 work to emphasize this: “Studies of genetic patterns in modern Nubians – both from the perspective of genome-wide data²⁰ and from the perspective of uniparental markers like mtDNA (added ref. Osman et al. 2021) – are thus important for informing on present-day populations, but our results show that these results cannot be simply extrapolated back to ancient Nubians. Instead, this requires ancient DNA data, such as we report on here.”

*In this manuscript, the finding that West Eurasian-related ancestry in Kulubnarti disproportionately derived from female ancestors is **not** based on uniparental haplogroups; indeed, we recognize and explicitly state that it is not possible to associate geographic origins based on uniparental haplogroups alone in this context because of the distribution of these haplogroups in both Eurasia and Africa, writing: “Examining uniparentally inherited parts of the genome (Methods; Supplementary Data 12; Supplementary Note 5; Supplementary Fig. 6), we find that 35 out of 63 individuals from both cemeteries who were not first-degree relatives sharing a maternal lineage belong to 11 mitochondrial DNA (mtDNA) haplogroups that are presently distributed predominantly in West Eurasia although the presence of such lineages for thousands of years in northeastern Africa as well has been established by previous work^{31,64,65}.”*

Instead, our assessment that West Eurasian-related ancestry at Kulubnarti is associated with females is based on comparison of ancestry proportions on the autosomes and on the X chromosome. This is a well-established method for exploring sex-biased admixture (see, for example, Mathieson et al. Nature 2018), and our results clearly indicate the overrepresentation of West Eurasian-related ancestry on the X chromosome relative to the autosomes in the Kulubnarti Nubians. To clarify our approach and to highlight the points the referee makes with which we agree, we have added the following sentence: “Previous morphological analysis has found no evidence of sex-specific patterns of mobility at Kulubnarti⁶⁰, while analysis of Y chromosome and mitochondrial DNA (mtDNA) are unlikely to paint a clear picture of sex-biased ancestry because many haplogroups that are common in West Eurasia are also found in northeastern Africa; however, genome-wide data provide a potentially more powerful way to investigate sex-biased ancestry.”

For example haplogroup H, used to justify major female gene flow in this manuscript (irrespective of indicators like culture or language), was shown to exist among Hausa and Fulani, two groups that shared recent history in the Sahel (Osman et al., 2021). Nubian influx itself is being proposed by archeologists to have begun from west of the Nile at some time three millennia ago.

We do not use the identification of individuals belonging to mtDNA haplogroup H to argue for female gene flow (see above: the signal of sex-biased gene flow is based on comparisons of ancestry proportions on the autosomes and X chromosome; uniparental haplogroups are reported in this work but are not definitive with regard to sex-biased gene flow as the confidence intervals are too noisy).

While the mtDNA H lineage may be present among present-day Hausa and Fulani, we note that the haplogroup H2a (primarily distributed in Europe) has “not previously [been] found in ancient contexts in Africa to our knowledge.” We draw no further conclusions about gene flow from this finding.

Assigning populations

Several terms were used to assign populations. Using the term Nilotic to denote Nilo-Saharan populations is a mistake, as Nilotics has been confined to a branch of Nilo-Saharan speakers mainly in southern Sudan, Ethiopia and the great lakes.

We thank the referee for highlighting this subtle distinction which was not as clear as it could have been in our manuscript. Here we use the term “Nilotic” to refer not to all Nilo-Saharan speakers but instead to Nilotic-speakers in specific parts of northeastern Africa, as the referee suggests. In particular, we are referring to Dinka, who can accurately be identified as a Nilotic group. To address the ambiguity highlighted in the referee’s comment, we have updated our definition to now read: “(in what follows, we use “Nilotic” to refer to the ancestry related to the people indigenous to parts of northeastern Africa, including southern parts of Sudan, who speak Nilo-Saharan languages; we emphasize Nilo-Saharan languages are spoken over a broader region, and in this paper we do not use the term “Nilotic” to refer to Nilo-Saharan speakers outside this core region).”

Similarly is the term Sudanese Arab. Arabic became the lingua franca in Sudan for at least a century for some time, the term Arab is uninformative of Ethnicity if not deceiving. Even “Arabic speaking” which is descriptive of cultural communities should include the name of the people/ethnic group.

When we twice use the term “Sudanese Arab” in the manuscript we are referring to the PCA, which utilizes data from Hollfelder et al. PLoS Genetics 2017 and we are replicating the nomenclature from that study. There is no further discussion in our manuscript of Sudanese Arab groups. To head off misunderstanding, we have caveated the use of the term in the following revised sentence: “A second cline correlates to increasing proportions of West Eurasian-related ancestry, extending from Nilo-Saharan-speakers to West Eurasians. Sudanese Arab (here we use a group identifier based on ethnic and linguistic categories following the original publication that reported the data), Beja, and Nubian people from the northeastern and central regions of Sudan, along with Afro-Asiatic-speakers from Ethiopia and Somalia, fall intermediate along this cline.”

Ethics and community engagement:

The authors utilize a rather outdated approval and consent that dates back to the "Campaign to Save the Monuments of Nubia". "Prior to the excavation of the Kulubnarti cemeteries, the head of the Sudanese Antiquities Service approved at the time ", something that is not compatible with present day good practices in ethics and research which include feeding back results to community, stewardship etc.... individuals' Perception of research aspects of identity and history can be quite dynamic. Also community engagement should not be confined to those isolated vulnerable communities in the site where research and excavation is carried out. Broader consent and consultation, which should extend to include the natural stewards and intellectual guardians of the heritage of these communities, such as local academic and educational institutions, no matter how poor their technological setting or academic tradition. This is not essential only for “political correctness” but for reasons that pertain, in addition to ethics, to the scientific rigor of the content, the representativeness of historical narratives and validity of conclusions in this case. Genuine and actual contributions should be the criteria leading to authorship or acknowledgement for both local and international scientists alike rather than token-like “piggybacking “ attributions.

We agree with the referee that genuine and actual contributions should be the criteria leading to authorship for all scholars (both local and international). It is precisely for this reason that we do not include any co-authors on this paper other than those individuals who made substantial intellectual contributions to this work (which started based on the co-senior author (Van Gerven’s) archaeological and physical and anthropology work which began decades ago, and subsequently grew more proximally from the first author’s Ph.D. dissertation work). We agree with the referee that it would have been tokenistic to add local

or international authors who did not make an intellectual contribution at a later stage for the sake of optics. We look forward to more extensive engagement with our local Sudanese collaborators on our ongoing work in the Nile Valley that has been seeded by the present work and that we fully expect will be reflected in major contribution from local scholars recognized by authorship in upcoming manuscripts.

We believe that while the approval and consent for this work was given many decades ago, the process of official approval and community engagement were consistent not only with the ethical best practices of the time, with present-day good practices that a globally diverse community of >60 scholars from >20 countries (including a participant from Sudan) agreed to in a November 4-5, 2020 online workshop that resulted in a manuscript currently in revision highlighting these recommendations.:

- (1) Researchers must ensure that all regulations were followed in the places where they work and from which the human remains derived. Our manuscript describes the formal permissions we obtained for the export and scientific analysis of the Kulubnarti Nubians. While these permissions were obtained years ago, they still represent the appropriate and necessary permissions to carry out this work.
- (2) Researchers must prepare a detailed plan prior to beginning any study. This work which represents the Ph.D. thesis of first author Kendra Sirak was funded by the National Science Foundation (Grant No. BCS-1613577), and in this context required the submission of a detailed research plan.
- (3) Researchers must minimize damage to human remains. Analysis of the Kulubnarti skeletal collection involved preferentially selecting individuals with accessible petrous bones that were disarticulated from the rest of the skull to maintain as many complete skulls as possible for any future analyses.
- (4) Researchers must ensure that data are made available following publication to allow critical reexamination of scientific findings. The paper commits to placing all aligned genomic sequences will be deposited upon publication into the European Nucleotide Archive (accession number PRJEB42975).
- (5) Researchers must engage with other stakeholders from the beginning of a study and ensure respect and sensitivity to stakeholder perspectives. The members of the community who lived in the village neighboring the Kulubnarti cemeteries and who used the mainland cemetery for the burial of their own deceased consented to the research following conversations with Dr. Dennis Van Gerven (co-senior author on this manuscript) before and during excavation. We are confident that from a cultural historical perspective there are no living Nubians more closely linked to the Christians we excavated than the current residents of Kulb who willingly permitted and participated in the excavation.

The reviewer notes that additional important activities can include “feeding back results to community” and involving institutions “such as local academic and educational institutions.” We agree this is an important part of ancient DNA work. Indeed, the first author shared this manuscript with a Sudanese archaeologist colleague from the University of Khartoum in February 2021 to obtain feedback about the content of the manuscript and obtain a local perspective on the way that results were discussed in terms of present-day notions of Sudanese identity. In line with the requests of this referee, the first author has also committed to returning results to Sudanese colleagues and communities and has developed a plan for this that has already been written into active research proposals. This plan includes:

1. The first author will give a lecture and a day-long workshop on the “What’s, how’s, and why’s of ancient DNA research” that will include a presentation of the Kulubnarti results to students of archaeology and anthropology at the University of Khartoum;

2. *The first author will give a lecture and a half-day-long workshop on the “What’s, how’s, and why’s of ancient DNA research” that will include a presentation of the Kulubnarti results to personnel at NCAM;*
3. *The first author will give a talk to the general public about ancient DNA research and hold a discussion session where questions and concerns about the research can be voiced; we are planning to hold this event at the British Council in Khartoum (venue to be confirmed in early fall);*
4. *The first author will provide training to collaborator Dr. Mohamed Bashir (University of Khartoum) and his colleagues on methods for sampling skeletal material for ancient DNA research; this capacity building activity comprises an important part of their collaborative National Geographic project. Training will how to carry out minimally-invasive method for sampling intact skulls that the first author published in Sirak et al. 2017. The first author will provide Dr. Bashir with the necessary equipment to carry out sampling independently in the future, ensuring that our collaborative work also generates new opportunities for him and his colleagues to expand their role in ancient DNA research in future projects.*
5. *The first author will create a poster detailing the results from the Kulubnarti project that will be shared with NCAM personnel for placement in the Sudan National Museum if desired.*

We are finally grateful to have had the perspectives of a local person introduced during the process of peer review at Nature Communications. We believe that the review process is one forum for engagement as well, and we appreciate the suggestions made by this reviewer that improved our manuscript.

REVIEWERS' COMMENTS

Reviewer #5 (Remarks to the Author):

I have read the response of Sirak et al. and appreciate their diligent response to various comments and remarks

In fact I was not expecting a rebuttal to all raised points some of which were made rather to evoke discussions and alert authors to gaps misconceptions and blank pages in the history of Nubia and Sudan. Having such contested issues is largely expected with the dearth of data and scarcity of literature from the region and its populations.

Current and future research with genome wide data informing history is increasingly bound to correct some of the inherent myths and misinformation on the history of our kind and this part of Africa, the paper by Sirak et al., is hopefully another milestone along this road.